



# Seasonal and regional variations of sinking in the subpolar North Atlantic from a high-resolution ocean model

Juan-Manuel Sayol[1], Henk Dijkstra[2], and Caroline Katsman[1]

[1]Department of Hydraulic Engineering, Delft University of Technology, Delft, The Netherlands

[2]Institute for Marine and Atmospheric research Utrecht, Utrecht University, Utrecht, The Netherlands

**Correspondence:** Juan-Manuel Sayol (J.M.SayolEspana@tudelft.nl)

**Abstract.** Previous studies have indicated that most of the net sinking associated with the downward branch of the Atlantic
Meridional Overturning Circulation (AMOC) must occur near the subpolar North Atlantic boundaries. In this work we have
used monthly mean fields of a high-resolution ocean model (0.1 deg at the equator) to quantify this sinking. To this end we have
calculated the Eulerian net vertical transport ($W_\Sigma$) from the modelled vertical velocities, its seasonal variability and its spatial
distribution under repeated climatological atmospheric forcing conditions. Based on this simulation, we find that for the whole
subpolar North Atlantic $W_\Sigma$ peaks at about $-14$ Sv at a depth of 1139 m, matching both the mean depth and the magnitude
of the meridional transport of the AMOC at $45^\circ$N. It displays a seasonal variability of around 10 Sv. Three sinking regimes
are identified according to the characteristics of the accumulated $W_\Sigma$ with respect to the distance to the coast: one within the
first 110 km and onto the bathymetric slope at around the peak of the boundary current speed (regime **I**), the second between
110 km and 290 km covering the remainder of the shelf where mesoscale eddies exchange properties (momentum, heat, mass)
between the interior and the boundary (regime **II**), and the third sinking regime at larger distances from the coast where $W_\Sigma$ is
mostly driven by the ocean's interior eddies (regime **III**). Regimes **I** and **II** accumulate $\sim 90\%$ of the total sinking and display
smaller seasonal changes and spatial variability than regime **III**. We find that such a distinction in regimes is also useful to
describe the characteristics of $W_\Sigma$ in marginal seas located far from the overflow areas, although the regime boundaries can
shift a few tens of km inshore or offshore depending on the bathymetric slope and shelf width of each marginal sea. The largest
contributions to the sinking come from the Labrador Sea, the Newfoundland region and the overflow regions. The magnitude,
the seasonal variability and the depth at which $W_\Sigma$ peaks vary for each region, thus revealing a complex picture of sinking in
the subpolar North Atlantic.



## 1   Introduction

The Atlantic Meridional Overturning Circulation (AMOC) is a fundamental component of the Earth's climate system (Lozier, 2012; Buckley and Marshall, 2016). Over the last decades the traditional view of an ocean conveyor with an upper poleward current transporting warm waters to higher latitudes, and a downward branch with intermediate and deeper denser waters that originate in the regions of deep convection and move toward the equator (Broecker, 1987, 1991) has been revised.

First, it became apparent that eddies actively mediate between the upper and lower limbs of the AMOC (Lozier, 2010). As a consequence, the Deep Western Boundary Current (DWBC) presents large spatio-temporal variability, as the flow splits in the North Atlantic (near $50°$ N) in the well-known western current aligned with the boundary (Stommel and Arons, 1959) and other more elusive interior paths (Bower and Hunt, 2000; Bower et al., 2009). Similarly, at the surface, the pathways of the North Atlantic Current that connect the subtropical gyre with the subpolar gyre are still not well established, as evidenced by the trajectories of surface drifters (Brambilla and Talley, 2006) and by estimates of the inter-gyre exchange (Rypina et al., 2011).

Second, the earlier idea that strong open ocean convection -e.g. in the interior of the Labrador, Irminger or Greenland Seas- is accompanied by large-scale sinking of waters, and that this downwelling represents the largest part of the sinking related to the AMOC, has been abandoned. The explanation for this is that once the winter cooling has preconditioned the convection site through a prolonged buoyancy loss, the vertical transport associated with small-scale ($\sim 1\,km$) deep convective plumes is mostly compensated by nearby rise of waters, so that little vertical mass transport is expected (Marshall and Schott, 1999; Send and Marshall, 1995). Moreover, these regions of deep convection are highly localized, with length scales of $500\,km$ of less. This implies that the horizontal gradient in planetary vorticity across the convection region is small. In order to balance the vorticity changes associated with substantial sinking in the geostrophic ocean interior, an unrealistically strong northward current would be required (Spall and Pickart, 2001). Instead, previous studies have shown that the Eulerian net sinking (in depth space) associated with the lower branch of the AMOC must occur near the boundaries, where the flow is subject to non-geostrophic dynamics. Thus, at the topographic slopes a richer vorticity balance arises with a dissipation term that compensates the vertical stretching of planetary vorticity induced by the sinking (Spall and Pickart, 2001; Spall, 2004, 2008; Brüggemann et al., 2017). As a result, higher rates of sinking are attainable near the boundaries of marginal seas.

Spall and Pickart (2001) and Pedlosky and Spall (2005) analyzed the sinking along a straight boundary current subject to buoyancy loss. They found that a flow in thermal-wind balance develops, with a density gradient that spirals with depth. Such a spiraling structure induces strong vertical movements, which they found to be proportional to the along-shore density gradient and the mixed layer depth. With a 2-layer approach, Straneo (2006) studied a boundary current surrounding denser interior waters as a representation of the Irminger Current waters flowing around the perimeter of the Labrador Sea. In her model net sinking appears in the boundary layer as the boundary current looses buoyancy along the perimeter. She also found that larger along-shore density gradients give rise to more sinking. Consistently Cenedese (2012) came to a similar conclusion from laboratory experiments in a tank. However, the North Atlantic is not the only place where sinking predominantly takes place near the boundaries. As pointed out by the recent work of Waldman et al. (2018), significant sinking occurs in the first 50 km off



the coast in the Mediterranean Sea, though it is much smaller than in the North Atlantic ($\sim -1$ Sv, where $1\,\mathrm{Sv} = 10^6\,\mathrm{m}^3\,\mathrm{s}^{-1}$).
At this location, sinking is catalyzed by the existence of a western boundary current that densifies along its way around the
basin, a strong winter cooling in the interior due to northerly winds, and an active near-shelf eddy field.
More recently, Katsman et al. (2018) estimated the net sinking in the North Atlantic from model simulations at a depth chosen
to match the maximum of the overturning streamfunction (at 1060 m), well below the mixed layer depth. They computed the
net sinking over the time averaged fields of two hindcasts based on the same ocean model (ORCA) and atmospheric forcing.
The two simulations covered the period 1958-2001 and had a different horizontal resolution ($0.25^{\circ}$ and $1^{\circ}$, —ORCA025 and
ORCA1— respectively). Their results showed a significant net sinking along the boundaries, much higher than in the interior.
Notably the finer resolution model displayed 8 Sv more net sinking along the perimeter than the lower resolution version
($-20$ Sv and $-12$ Sv respectively). However, the contribution of overflow waters to the total budget of net sinking along the
selected perimeter was nearly the same in absolute terms, with average amounts of $-7.6$ Sv and $-7.4$ Sv for the coarser and the
finer resolution simulation respectively. Hence, the large differences in sinking between both simulations are mostly attributed
to the boundary region. According to the authors, this difference may be due to the fact that the finer resolution model is
eddy permitting. Thus, ageostrophic eddy-driven processes may also play an important role in boundary sinking. For instance,
eddy-induced heat fluxes significantly increase the lateral heat exchange between the cooler interior and the warmer boundary,
cooling the boundary current on its way and then enhance the along-shore density gradient (Spall, 2011). Also, as pointed out
by Spall (2010) eddy fluxes and dissipation play an important role in balancing the vertical stretching of planetary vorticity
induced by sinking in a convective basin.
In order to better understand the contribution of geostrophic and ageostrophic processes to sinking, Brüggemann et al. (2019)
used an idealized model with fine resolution (3 km in the horizontal), which is able to mimic the basic features of the Labrador
Sea: a cyclonic boundary current circulating along a semicircular basin, with a small part dominated by a steeper topographic
slope (change in depth of 3000 m in few tens of km) resulting in the generation of a vigorous eddy field. Stronger sinking
was found onto and near the sharp topographic feature than in any other place in the domain. To quantify the importance of
all involved processes in the amount of net sinking the authors decomposed the vorticity balance separating mean from eddy
terms; it included the stretching of planetary vorticity, the mean and eddy horizontal advection of vorticity as well as the $\beta$
and dissipation terms. One important result is that the intense vertical stretching of planetary vorticity associated with the
sinking is balanced by the horizontal advection of vorticity and dissipation. Likewise, the recent work of Georgiou et al. (2019)
highlights the importance of eddy-driven transport using and idealized eddy-resolving model of the Labrador basin. Among
others, this study demonstrates that the total amount of sinking is sensitive to changes in the eddy pathways. It must be stressed
here that the above idealized studies assumed closed basins, while in reality open boundaries exist and exchanges between the
North Atlantic and Arctic occur. Overflows also contribute significantly to the net sinking, as shown in Katsman et al. (2018).
However, they are governed by a different dynamics (Shapiro and Hill, 1997; Yankovsky and Legg, 2019).
This work adds a new dimension of complexity to existing studies by investigating how the net sinking in the North At-
lantic changes seasonally and regionally. Despite the promising first results of the Overturning in the Subpolar North Atlantic
Program-OSNAP (Lozier et al., 2017; Kornei, 2018; Holliday et al., 2018; Lozier et al., 2019), the scarcity of measurements





below the surface still necessitates the use of numerical models to provide more insight into the spatio-temporal variability of
sinking and to grasp the physical processes behind its dynamics. To this end, we use an ocean-only eddy-resolving numerical
simulation with a nominal resolution of $0.1°$ under a repeated climatological annual atmospheric forcing, not adding even
more complexity. Since the degree of buoyancy loss, the topographic configuration, and the oceanic circulation differ among
the North Atlantic sub-basins, a complex repartitioning of sinking is anticipated. Therefore we evaluate separately the season-
ality and the distribution of sinking at different spatial scales including the marginal seas and overflow areas in the basin. In
addition, the connection between sinking and the AMOC is addressed throughout the paper.
The remainder of this paper is structured as follows: in Sect. 2 we introduce the numerical simulation and assess the ability
of the model to reproduce a realistic AMOC; in Sect. 3 we consider the main characteristics and the seasonal variability of
sinking in the entire subpolar North Atlantic; in Sect. 4 we evaluate similarities and differences between sinking in the marginal
seas, overflow regions and mid-latitude seas of the subpolar North Atlantic based on their different bathymetric profiles and
driving local dynamical processes; in Sect. 5 we show that in our simulation the connection between sinking variations and the
AMOC changes fades when the dominant seasonal signal is removed. To conclude, in Sect. 6 we summarize and discuss the
most significant findings.

## 2   Model data & methods

The model set up is based on a configuration of the Parallel Ocean Program model (POP) (Maltrud et al., 2008; Smith et al.,
2010). This model solves the primitive equations on a tri-polar curvilinear grid with a nominal horizontal resolution of $0.1°$
at the equator and 42 $z$-layers in the vertical down to a depth of 6000 m. The vertical resolution ranges from $10\,\mathrm{m}$ at the
surface to $250\,\mathrm{m}$ for the deepest layers. The bottom topography is described by partial bottom cells (Adcroft et al., 1997). The
atmospheric forcing fields (wind, heat fluxes and precipitation) are applied by repeating a prescribed annual cycle from the
Coordinated Ocean Reference Experiment (CORE) forcing dataset (Large and Yeager, 2004). Observed river run-off fields are
also included. Table 1 shows the value of some key model parameters. The simulation analyzed here is a subset of a longer
control run already employed by Brunnabend and Dijkstra (2017) — see the latter for more details on this simulation.

**Table 1.** Summary of POP model key parameters used in the simulation (see Maltrud et al. (2010); Weijer et al. (2012); Brunnabend and Dijkstra (2017) for details)

| Parameter | Value |
|---|---|
| Horizontal resolution | 0.1 deg at the equator |
| Vertical resolution | 42 non-equidistant $z$-levels. From 10 m (surface) to 250 m (deepest) |
| Horizontal dissipation (momentum) | Bi-harmonic viscosity and diffusion $\propto \mathrm{grid\,size}^3$. At the equator $\nu_0 = -90\,\mathrm{m}^4/\mathrm{s}$ |
| Horizontal dissipation (tracers) | Bi-harmonic viscosity and diffusion $\propto \mathrm{grid\,size}^3$. At the equator $k_0 = -30\,\mathrm{m}^4/\mathrm{s}$ |
| Vertical Mixing (K-profile) | $0.1\,\mathrm{m}^2/\mathrm{s}$ to solve gravitational instabilities |
| Background vertical tracer diffusion | From $10^{-5}\,\mathrm{m}^2/\mathrm{s}$ (surface) to $10^{-4}\,\mathrm{m}^2/\mathrm{s}$ (depth) |





## 2.1 Model data & general performance

We use 15 years of three-dimensional monthly-mean fields of velocity, potential density, temperature and salinity for our analysis. Other two-dimensional variables such as bottom depth, the area and volume of grid cells and monthly-mean fields of mixed layer depth are also utilized. Since this study focuses on the seasonal time scale, and because the forcing is annually repeated, 15 years is sufficient to provide robust results. The period selected, corresponding to years 260-274 in the simulation time frame, is chosen well after the spin-up years. We note that inter-annual and larger time scales of variability are not included in our prescribed repeated annual forcing. This does not alter the validity of our analysis at seasonal scales since the annual cycle of winds and heat fluxes are dominant. Maps with mean ocean currents for the North Atlantic Ocean at depths of 5 m and 1139 m (Fig. S1) show realistic strength and location of the Gulf Stream and other subpolar boundary currents, taking into account the resolution of the model. Previous work using the same model in a similar set-up found a well-represented distribution of currents, kinetic energy and water-mass properties at basin scale (Maltrud et al., 2010; Weijer et al., 2012; Brunnabend and Dijkstra, 2017). Moreover the modelled mixed layer depth qualitatively matches the spatial pattern derived from ARGO floats (Våge et al., 2009; Holte et al., 2017), where the areas of deepest convection are found in the south-west Labrador Sea, in the Greenland Sea and around the Iceland-Scotland Ridge. However the modelled data shows some delay in reaching the deepest mixed layer depth in the Labrador Sea (March against April) and tends to overestimate the observed mean values in some areas (compare Fig. S2 with Fig. S3). We partly attribute this to the use of repeated normal-year forcing conditions for wind and heat fluxes. We note also that the spatial coverage by ARGO floats is still scarce and hence gridded fields are coarser than model data. Besides, both results are sensitive to the algorithm used to compute the mixed layer depth (de Boyer Montégut et al., 2004).

## 2.2 Overturning streamfunction

The overturning streamfunction ($\psi_\mathrm{o}$) is a measure of the AMOC strength. With this metric northward and southward flows can be identified, as well as the depth where the transport reaches its maximum. $\psi_\mathrm{o}$ is determined from the vertical integral of the zonally integrated meridional velocity at the southern boundary of our domain, and the running meridional integral of the zonally integrated vertical velocity, i.e.:

$$\psi_\mathrm{o}(y,z,t) = -\int\limits_{x_w}^{x_e}\int\limits_{H(x',y')}^{z} v(x',y_0,z',t)\,dz'\,dx' + \int\limits_{y_0}^{y}\int\limits_{x_w}^{x_e} w(x',y',z',t)\,dx'\,dy' \tag{1}$$

where $v(x',y_0,z',t)$ and $w(x',y',z',t)$ are the meridional and vertical ocean velocity components, $y_0$ is latitude of the southern boundary (selected at $y_0 = 25°\mathrm{N}$), $H(x',y')$ is the ocean bottom depth and $x_w$ and $x_e$ are the western and eastern boundaries of the North Atlantic Ocean respectively (Fig. 1A). The model simulation analyzed here yields a maximum time-averaged overturning streamfunction of $25.6\,\mathrm{Sv}$ near $35°\mathrm{N}$, whereas the modelled $\psi_\mathrm{o}$ at the RAPID array location ($26°\mathrm{N}$) shows a maximum time-average transport of $22.3 \pm 1.9$ Sv (Fig. 1B, blue line). This value is within the interval of uncertainty





of the annual mean RAPID array observations prior to 2008, $18.8 \pm 4.3$ Sv (Cunningham et al., 2007; Kanzow et al., 2010),
and slightly larger than observations if we consider a longer RAPID period (April 2004-January 2017) with $17.0 \pm 1.9$ Sv (the
uncertainty is the standard deviation). Recent model-based results from Sinha et al. (2018) indicate that the RAPID array may
be underestimating the transport by about 1.5 Sv, due to structural errors in the array set-up. Not surprisingly, our simulation
is less successful in reproducing the range of variability of annual averages at the RAPID array (April 2004-January 2017),
underestimating this variability by almost 5 Sv, with a range of 3.2 Sv against the measured 7.9 Sv (Smeed et al., 2014, 2018).
This underestimation is likely due to the use of seasonal mean wind forcing conditions, where the atmospheric high-frequency
variability has been partially filtered. Finally, the depth of the maximum time-averaged $\psi_o$ is 1139 m (Fig. 1A), very close
to the depth found at the RAPID array location, at about 1100 m (Smeed et al., 2014, 2018). At $45°$ N (red line in Fig. 1B),
the modelled AMOC is around 8 Sv weaker than at the RAPID array location but presents a more pronounced seasonal cycle
(around 10 Sv), with the maximum in August and the minimum in February. Results from two dedicated campaigns in different
years covering the OSNAP sections ($\sim 50° - 60°$ N) provided a similar range of variability for $\psi_o$, $\sim 10 - 20$ Sv (Holliday et al.,
2018). The recent first assessment of the OSNAP observations by Lozier et al. (2019) yields a mean estimate of the $\psi_o$ that
is smaller than our mean $\psi_o$ at $45°$ N (around 6 Sv smaller) but that compares well with our mean modeled results at $55°$ N
($8.0 \pm 0.7$ Sv versus $8.1 \pm 2.4$ Sv, not shown). The depths of maximum $\psi_o$ are located at around 1000 m for the OSNAP
observations and at around 1100 m for our simulation for both $45°$ N and $55°$ N. In contrast, their results do not show a clear
seasonal signal in the AMOC while our simulation shows a marked seasonality in $\psi_o$ for $45°$ N ($\sim 10$ Sv) and $55°$ N ($\sim 8$ Sv).
We partly attribute this difference between the observations and our modeled results to the short OSNAP time series (only 21
months, from August 2014 to April 2016), and to the fact that we are using normal forcing conditions with a dominant seasonal
signal without high-frequency wind variability. Nevertheless, it can be inferred a stronger transport in summer than in winter
from the OSNAP observations (Lozier et al., 2019), which is in agreement with our modeled results. Therefore we conclude
that our simulation displays an AMOC with reasonable mean transport and variability, and a well-located core in depth.

## 3  Mean and seasonal characterization of net sinking in the subpolar North Atlantic

Thea AMOC only provides a two-dimensional view of the overturning circulation in the subpolar North Atlantic. In this study
we analyze the complex full structure of the circulation by characterizing spatial and seasonal variations in the sinking. In Sect.
3.1 we present the spatial distribution of modelled vertical velocities, which we use to compute the net vertical transport for the
subpolar North Atlantic Ocean, its seasonal variability and its vertical structure. In Sect. 3.2 a distinction in sinking regimes is
proposed based on the differences in the net vertical transport between the near-shelf and the interior regions. To conclude this
Section, we discuss our results in light of earlier studies.

### 3.1  Vertical structure of sinking

A map of the mean distribution of the maximum vertical velocities ($\overline{w}_{max}$, the sign is conserved) in the North Atlantic (Fig. 2A)
shows a spatial pattern characterized by strong velocities mostly confined to the boundaries. This is in line with results from



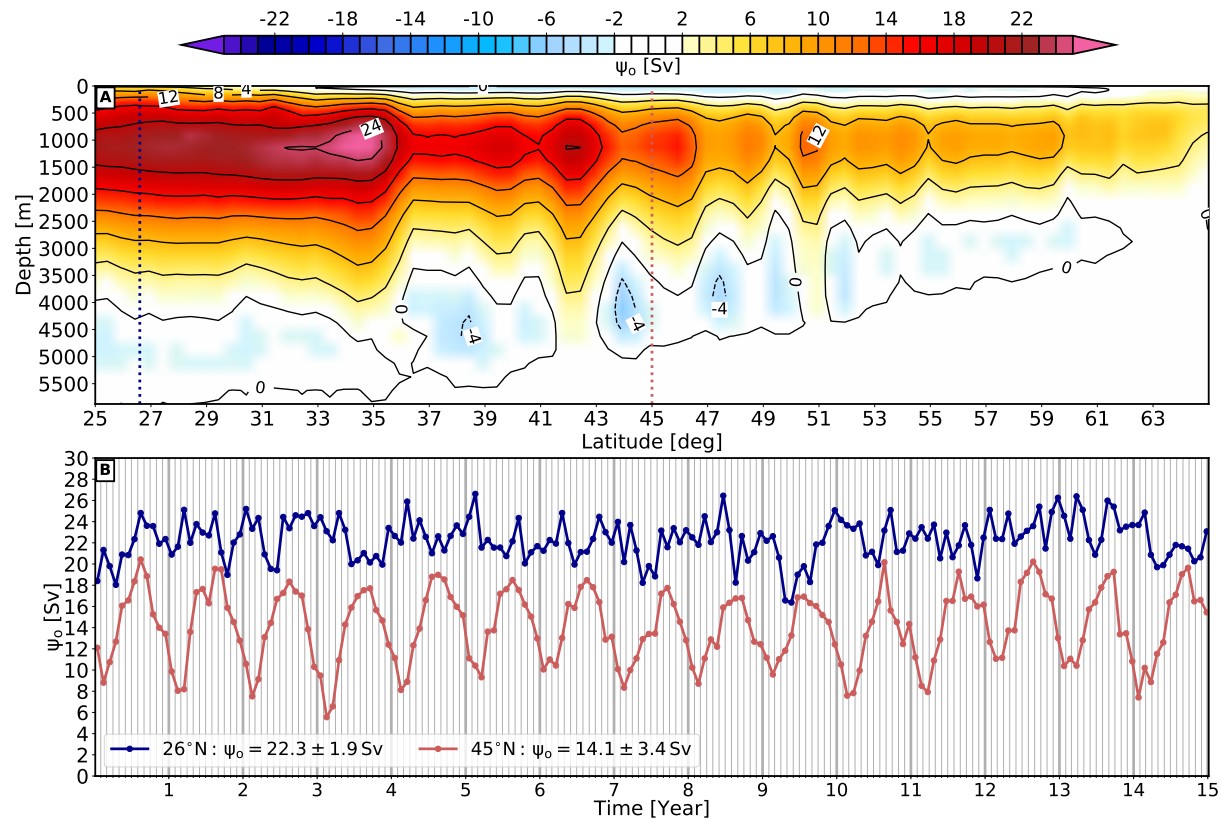

**Figure 1.** (A) 15-year average (years 260-274) overturning streamfunction, $\psi_o(y,z)$, for the North Atlantic Ocean. (B) Time series of maximum overturning streamfunction at 26°N (blue) and 45°N (red); positions are also indicated by dashed lines in Fig. 1A. $1\,\mathrm{Sv} = 10^6\,\mathrm{m}^3\,\mathrm{s}^{-1}$.

idealized models (Spall and Pickart, 2001; Spall, 2004; Straneo, 2006; Spall, 2010; Georgiou et al., 2019; Brüggemann et al.,
2019) and global ocean models (Katsman et al., 2018). $\overline{w}_{\max}$ may reach values of over $150\,\mathrm{m\,day}^{-1}$, in good agreement with
glider-based observations (Frajka-Williams et al., 2011). Fig. 2B shows the depth at which the velocities are most intense; the
black points mark where $|\overline{w}_{\max}|$ is larger than $80\,\mathrm{m\,day}^{-1}$. Most of the strong vertical velocities are found at a depth around
1000 meters (black contour in Fig. 2A) close to the boundaries. However, there are some exceptions, such as near the Flemish
Cap and in the interior of the Greenland and Norwegian Seas. There, strong vertical velocities are found farther offshore. At
some locations, alternating patterns of upward and downward motions are found (Fig. 2A, south-east of Greenland). Possibly,
water is forced to move up and down there due to the dynamical restrictions imposed by the full vorticity balance on topographic
slopes (see e.g., Spall, 2010; Brüggemann et al., 2019). The positive and negative alternation near the Flemish Cap needs to
have a different cause, such as eddy-induced vertical velocities. Indeed, the high variance of vertical velocities $\sigma^2(w)$ near the
Flemish Cap is a reflection of the existence of an active eddy field throughout the year (Fig. 2C). Also the subsurface EKE
shows this signal (Fig. S4). Note that the depth of $\overline{w}_{\max}$ in this region is much larger than 1000 m, which suggests the presence
of deep eddies with a strong barotropic component.



**Figure 2.** (A) 15-year maximum mean vertical velocity ($\overline{w}_{\mathrm{max}}$) for the North Atlantic Ocean. Contour lines denote the two longest 1000 m bathymetric features, which are separated by the Denmark Strait and the Iceland-Scotland ridge. (B) Depth of $\overline{w}_{\mathrm{max}}$ plotted in (A). Black dots mark those grid cells where $|\overline{w}_{\mathrm{max}}|$ is larger than 80 m day$^{-1}$. (C) 15-year variance of vertical velocity ($\sigma^2(w)$) at a depth of 1139 m. This depth corresponds to the depth at which the vertical transport associated with the AMOC peaks.



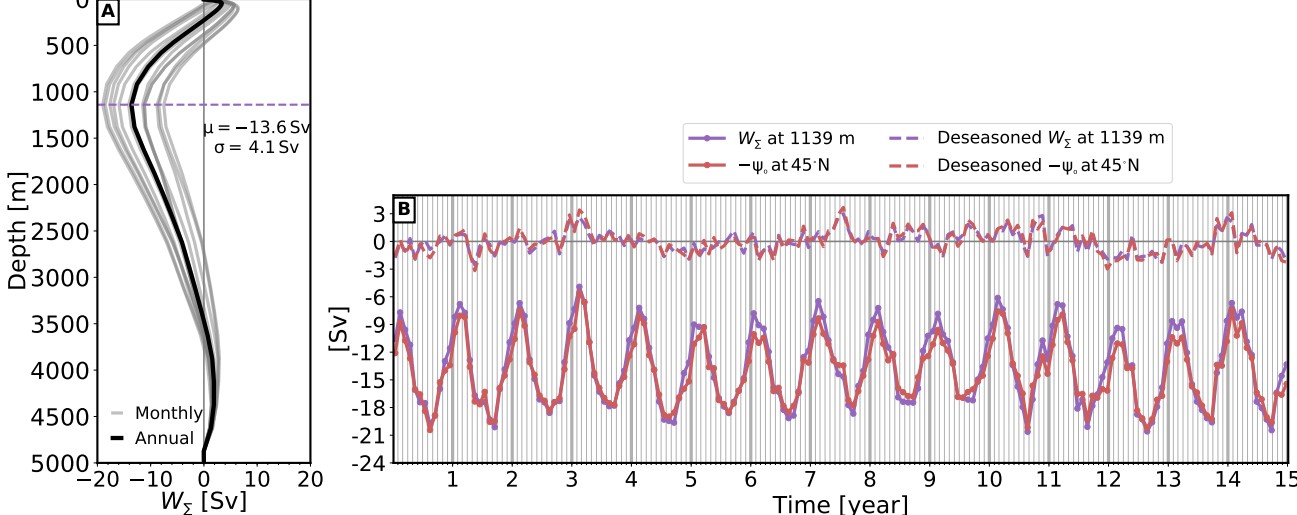

**Figure 3.** (A) Mean profile of the net vertical transport ($W_\Sigma$) for the region of study [$66°\,W - 20°\,E,\ 45°\,N - 75°\,N$]. The annual mean profile is shown by a thick black line; as the monthly climatology is indicated by gray lines. Mean and standard deviation ($\mu, \sigma$) are given in the legend; the depth of largest sinking (1139 m) is indicated with a horizontal purple line. (B) Time series of $W_\Sigma$ at 1139 m (purple lines, in Sv) compared to the reversed time series of maximum $\psi_o$ (Figure 1B) at $45°\,N$ (red lines, in Sv). Solid lines include the seasonality while dashed lines do not include the seasonal signal (see legend). The Pearson correlation coefficient between both time series is over 0.9 for both cases.

To assess the magnitude and the depth at which the near-boundary sinking occurs, we sum the local vertical transport for
the entire subpolar North Atlantic. First, we calculate the vertical transport for all model grid points and for every depth
as $W(x,y,z,t) = w(x,y,z,t)A(x,y)$, where $A(x,y)$ is the area of the grid cell, which depends on its location $(x,y)$ in our
curvilinear grid. Second, we sum $W$ over the horizontal domain of study defined in Fig. 2. We will refer to this net vertical
transport as $W_\Sigma$ for simplicity. The vertical profile of $W_\Sigma$ is shown in Fig. 3A. Large negative values of $W_\Sigma$ are found between
500-2700 m, with the strongest downward transport located at a depth of 1139 m. By mass conservation we expect that $W_\Sigma$
in our domain will be closely related to the transport at the southern boundary of the domain (45°N) since the North Atlantic
Current is the dominant feature in the basin, although some mass contribution from the Arctic Ocean at 75°N and through the
Davis Strait can be expected (Rudels et al., 2005; Azetsu-Scott et al., 2012). A comparison between time series of minimum
$W_\Sigma$ and the maximum $\psi_o$ (Fig. 1B) yields an excellent agreement in magnitude: $14.1 \pm 3.4\,\mathrm{Sv}$ against $-13.6 \pm 4.1\,\mathrm{Sv}$ (Fig. 1B
and Fig. 3A). If we compare the reversed time series of $\psi_o$ at $45°\,N$ (solid red line in Fig. 3B) with the time series of $W_\Sigma$ at the
depth of minimum $W_\Sigma$ (solid purple line) it is clear that also the seasonal signal matches, with the minimum $W_\Sigma$ in summer
(August) and the maximum in winter (February). The broadest range of variability (maximum minus minimum) is around 12
Sv in both time series at $\sim$1100 m depth. If we repeat the comparison after removing the seasonal signal from both time series
(dashed lines in Fig. 3B) the high correlation persists ($> 0.9$), but with a reduced maximum range of variability of 5 Sv.



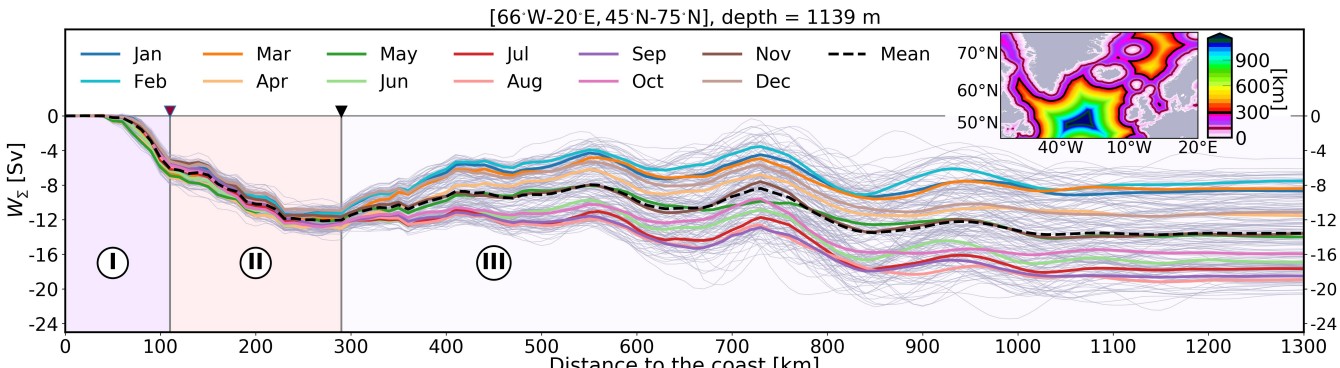

**Figure 4.** Cumulative net vertical transport ($W_\Sigma$) at a depth of 1139 m (in Sv) as a function of the distance from the coast defined in the inset map (in km). The dashed black line shows the annual mean. Monthly values of the 15-year simulation are shown in light gray, colored lines indicate the monthly climatology. The regimes of sinking are indicated by roman numbers **I-II-III**, and the separation lines between them are also denoted by a brown and a black triangle and by contours in the same color in the inset figure.

## 3.2 Variation of sinking according to distance from the coast

In order to quantify how much sinking takes place near the boundaries versus in the interior we have first classified all ocean grid points within the study area according to their distance to the nearest land point (inset map in Fig. 4); Next, we have accumulated $W$ starting from the coast towards the interior at the depth where $W_\Sigma$ (note the added spatial dependence) is at its minimum (at 1139 m, Fig. 3A). On average (dashed black line in Fig. 4), $-12$ of $-13.6$ Sv ($\sim 90\%$) of $W_\Sigma$ occurs in the first 290 km off the coast. A first assessment suggests the existence of three different sinking regimes according to the distance to the coast (indicated as regimes **I-II-III** in Fig. 4):

**I** *Distance* $\leq 110$ km. This region presents the largest increase in sinking with respect to the distance to the coast. It displays a small seasonal variation of less than 2 Sv. The minimum accumulated $W_\Sigma$ at 1139 m occurs in May (dark green line) and is around $-7$ Sv over a distance of $\sim 60$ km.

**II** *Distance between* $110 - 290$ km. The accumulated $W_\Sigma$ in this section remains intense though smaller than in regime **I** with $\sim -5$ Sv over 100 km. The magnitude of accumulated $W_\Sigma$ increases until about 220 km from the coast. Between 220 and 290 km the curve flattens, indicating that no additional sinking occurs or that it is locally compensated by rising waters. Seasonal variations are slightly larger than for the regime **I**, with values over 3 Sv between the months of April (orange line) and December (light brown line).

**III** *Distance* $> 290$ km. Beyond 290 km from the coast the trend in accumulated $W_\Sigma$ can revert completely with respect to regimes **I** and **II** depending on the season. During winter months, there is a net positive accumulated $W_\Sigma$ (i.e. net upwelling) between 290 and 750 km. During summer months a negative accumulated $W_\Sigma$ tends to occur. As a result, the final amount of accumulated $W_\Sigma$ displays a large seasonal variability, with deviations of up to 11 Sv between winter




and summer months at a distance of over 750 km from the coast. The annual mean of accumulated $W_\Sigma$ varies little with
distance from the coast in this regime (only $1 - 2$ Sv, compare the black dashed line in Fig. 4 at 290 km and at 1000 km).
The seasonal variations strongly affect the accumulated $W_\Sigma$ in some specific months (e.g. in February -light blue line-
or in August -pink line-), yielding changes in the accumulated $W_\Sigma$ of up to $50\%$ of what is seen in the first 290 km off
the coast.
It is hypothesized that these three sinking regimes reflect the effect of different processes contributing to the sinking. This is
illustrated in Fig. 5, by means of a time-mean vertical cross-section of the horizontal and vertical velocity field (see inset panel
A). First, there is sinking of waters between 500 and 2000 m along the Greenland shelf-slope all year round (black arrows
on the right hand of panels A-D). This sinking is connected with regimes **I** and **II** (Fig. 4), and occurs within the boundary
current (blue shading) below the mixed layer depth (light green line). Most of this sinking is constrained to a distance around
200 km off the coast in a region where isopycnals are tilted, and displays little seasonal variation. Second, there is a permanent
anticyclonic eddy of about 200 km of diameter at 1000-1250 km south of the tip of Greenland that extends from the surface to
a depth below 4000 m (shading), and generates both intense positive and negative vertical velocities at its flank (black arrows).
This pattern of interior eddy-induced rising/sinking of waters yields a small net annual mean vertical transport over the entire
basin but significant seasonal variability, which is reflected in sinking regime **III** (Fig. 4).
The boundary sinking found by Katsman et al. (2018) in a global ocean model and here characterized by regimes **I** and **II** is
captured by the ageostrophic theory and idealized models (Spall, 2010; Brüggemann et al., 2019; Georgiou et al., 2019), which
explained its basic features in terms of the vorticity balance. Thus, the narrow band of sinking closer to the coast represented
by the sinking regime **I** is characterized by the preeminent role of the topographically induced dissipation, while the sinking
farther offshore represented by regime **II** is largely driven by the presence of eddies near the boundary. Indeed, the amount
of sinking is governed by eddy-advection in the cross-shore direction (Georgiou et al., 2019), so it is not surprising that this
region presents a larger seasonal signal compared to the one described by regime **I** (Fig. 4, see also the patches of strong EKE
near the southern tip of Greenland in Fig. S4).
Spall and Pickart (2001) derived a simple expression to estimate the magnitude of meridional overturning ($M_B$) —or by
mass conservation, the downward vertical transport $W_\Sigma$— near the boundary for a situation with a deep mixed layer:

$$M_B = W_\Sigma = \frac{g\Delta\rho_B h^2}{2\rho_0 f}, \tag{2}$$

where the amount of $W_\Sigma$ is proportional to the square of the mixed layer depth ($h$) and to along-shore differences in potential
density ($\Delta\rho_B$) —$g$ is the Earth's surface gravity, $f$ the Coriolis parameter and $\rho_0$ a reference density—. Although Brüggemann
et al. (2019) have shown that Eq. (2) is not formally correct when the mixed layer depth is shallow (as is the case here), Katsman
et al. (2018) demonstrated that the relationship yields reasonable results in a realistic global ocean model when the mixed layer
depth ($h$) is substituted by half of the depth of largest sinking and the along-shore density change ($\Delta\rho_B$, which for this situation
depends on depth) by its depth average. The Eq. (2) indicates that the net vertical transport is among others controlled by the





**Figure 5.** 15-year climatology of the velocity field at a cross-section between the southern tip of Greenland and the southern limit of the study area (see inset in panel A). Each panel represents a seasonal average: (A) JFM (January-February-March); (B) AMJ (April-May-June); (C) JAS (July-August-September); OND (October-November-December). The shading shows the $u$ component of velocity (units in $\mathrm{m\,s^{-1}}$); black arrows are velocity vectors constructed as $(v, 1000 \cdot w)$. For clarity arrows are shown for 1 of every 3 horizontal grid points at certain depths. The green line depicts the seasonal mean mixed layer depth (in m), the black contours denote the seasonal potential density anomaly, $\sigma_\rho = \rho - 1000$, for selected values. The limits of the proposed sinking regimes (**I-II-III**) are sketched in panel B through vertical dark red dashed lines.



local along-shore density gradient ($\Delta\rho_B$), that is, a negative $W_\Sigma$ is associated with a rise in the isopycnals along the boundary
current (or equivalently, by the densification of waters at a given depth). This proportionality of the boundary sinking to the
density gradient along the boundary was pointed out by Straneo (2006) in her two-layer model approach. This connection
is also suggested here by the strong vertical velocities in the boundaries (Fig. 2A-B) and by the upward displacement of the
isopycnals between the eastern (south-east of Iceland) and the western (south-west of Greenland) sides of the basin (see the
mean depth of isopycnals of $\sigma_\rho = 27.75$ and $27.8\ \mathrm{kg \cdot m^{-3}}$ in Fig. S5A-B). Indeed this along-shore tilting is always present
and has its maximum in spring (Fig. S6).
Apart from the isopycnal tilting, Brüggemann et al. (2019) found that the cross-shore density gradients also contribute to the
budget of boundary sinking since, as eddies arise from baroclinic instabilities, they try to flatten the isopycnals. This can be
accompanied by strong vertical velocities and, hence, more sinking (see for example the cross-shore density gradient in Fig. 5,
where the isopycnal of $\sigma_\rho = 27.8\ \mathrm{kg \cdot m^{-3}}$ is tilted within the boundary current near the southern tip of Greenland).
Finally, the sinking in regime **III** is related to those processes that develop away from the shelf and far from the core of the
boundary current. Therefore eddies are more likely to drive this series of positive and negative events of vertical transport (Fig.
2A and Fig. 5). The major role of such quasi-permanent eddies is supported by the marked fluctuations between 300 and 1000
km in Fig. 4, the large interior eddy in Fig. 5 and the vigorous EKE field around the Flemish Cap (Fig. S4).

## 4   Regional distribution of net vertical transport

The overall view of net vertical transport ($W_\Sigma$) in the subpolar North Atlantic presented in the Sect. 3 may not be valid
at regional scales, since the bathymetric configuration, the ocean circulation and water-mass properties differ between the
different seas. Moreover, the dynamics of overflows are different from those governing the near-boundary sinking induced by
the buoyancy loss of a boundary current. In order to assess and understand these expected spatial variations we divide the
subpolar North Atlantic in eight well-established regions (see Fig. 6), which for discussion purposes can be grouped into three
more general sets: marginal seas (Labrador, Irminger, Greenland and Norwegian Seas -Sect. 4.1-), overflow regions (Denmark
Strait and Iceland-Scotland Ridge -Sect. 4.2-) and mid-latitude seas (Newfoundland and Rockall, -Sect. 4.3-). In the remainder
of this Section we will describe the following properties associated to $W_\Sigma$ for these three groups of regions:
(i) The time-mean $W_\Sigma$ at the depth of minimum $W_\Sigma$ (hereinafter this depth is defined as $z_{\min}$).
(ii) The seasonal variability of $W_\Sigma$ and $z_{\min}$.
(iii) The signal to noise ratio (SNR), defined as $\mathrm{SNR} = \left|\frac{\mu}{\sigma}\right|$ ($\mu$ = time-mean of $W_\Sigma$ at $z_{\min}$, $\sigma$ = standard deviation of $W_\Sigma$
at $z_{\min}$), a high value (SNR > 1) indicates that $\mu$ is relatively large compared to $\sigma$, whereas a small value (SNR < 1)
denotes a signal with a large temporal variability compared to the mean—although it does not necessarily imply a well-
defined seasonal signal as SNR does not yield any information on its periodicity—.
(iv) The regimes of sinking that can be identified.





In Sect. 4.4, a further evaluation of the sinking characteristics Labrador Sea, Irminger Sea and Newfoundland regions (which
represent about $2/3$ of net sinking in the subpolar North Atlantic and cover all three types of regions) is provided.

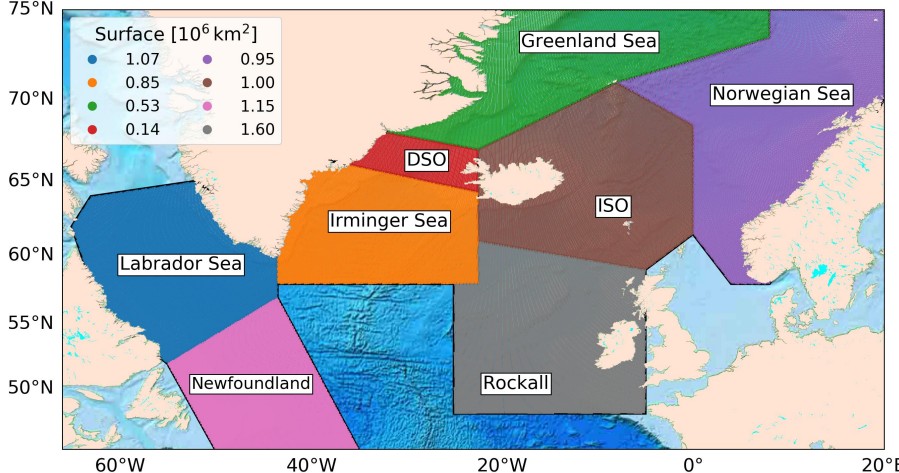

**Figure 6.** Map of the North Atlantic $[66°\,\mathrm{W} - 20°\,\mathrm{E}, 45°\,\mathrm{N} - 75°\,\mathrm{N}]$ divided into eight regions. DSO and ISO refer to Denmark Strait and Iceland-Scotland Ridge overflow regions respectively. The surface area of each region is shown in the legend in $10^6 \cdot \mathrm{km}^2$.

### 4.1   Marginal seas

Vertical profiles of $W_\Sigma$ for the marginal seas (Fig. 7A-D) indicate that on average the Labrador Sea contributes about twice as
much to the sinking as the other three marginal seas combined ($-4.04$ Sv against $-2.1$ Sv), yielding a total mean $W_\Sigma$ of $-6.14$
Sv in the marginal seas (see $\mu$ in Fig. 7A-D, and Table 2 for a complete summary; notice the different value of $z_{\min}$ for each
region). This contribution from the Labrador Sea is larger than the $-1.4$ Sv derived by Katsman et al. (2018) using a coarser
ocean model (ORCA025). This is probably due to the improved ability of higher-resolution models to resolve the eddy activity
and ageostrophic processes near the boundary (Georgiou et al., 2019; Brüggemann et al., 2019), which gives rise to stronger
vertical transports. It is also larger than the $-1$ Sv estimated by Pickart and Spall (2007) using the World Ocean Circulation
Experiment (WOCE) AR7W line data and the $-1.2$ Sv estimated from Argo floats by Holte and Straneo (2017) at a shallower
depth (around 800 m). This substantial difference may be due to the scarcity of observations. Interestingly the value of $z_{\min}$ for
the Labrador Sea matches the one previously shown in Fig. 3A for the whole basin (1139 m). The Greenland Sea also shows a
time-mean $W_\Sigma$ that stays negative during the whole year of about $-1.1$ Sv (Fig. 7C), with the minimum $W_\Sigma$ (largest sinking)
occurring in June and the maximum in February (Table 2). It displays a similar vertical shape as in the Labrador Sea, although
$z_{\min}$ is now located at a depth near 730 m. The sinking in the Irminger and Norwegian Seas is more variable and depends on the
time of the year yielding net positive (negative) $W_\Sigma$ during winter (summer) months (Fig. 7B,D). This yields a smaller annual
mean sinking than in the Labrador and Greenland Seas of -0.73 Sv and -0.25 Sv respectively. A notable difference between the

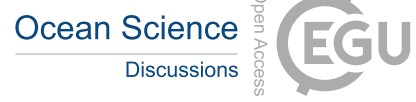

Irminger and Norwegian Seas is the value of $z_{\min}$, which is $\sim 470$ m for the Norwegian Sea and $\sim 1630$ m for the Irminger
Sea.

**Table 2.** Summary of properties of the sinking shown in Fig. 3A and Fig. 7 for the entire study area (Domain) and all regions defined in Fig. 6. $\mu$ denotes the time-mean net vertical transport ($W_\Sigma$) at the depth of minimum $W_\Sigma$ ($z_{\min}$), and $\sigma$ its the standard deviation. Min and Max denote the months when the minimum and the maximum $W_\Sigma$ (or equivalently the largest and the smallest sinking) occur respectively. The final column shows the signal to noise ratio (SNR), defined as $\mathrm{SNR} = \left| \frac{\mu}{\sigma} \right|$.

| Area | $\mu \pm \sigma$ **[Sv]** | $z_{\min}$ | **Min** | **Max** | $\mathrm{SNR} = \left\| \frac{\mu}{\sigma} \right\|$ |
|---|---|---|---|---|---|
| Domain | $-13.6 \pm 4.1$ | 1139 | August | February | 3.3 |
| Labrador Sea | $-4.04 \pm 0.83$ | 1139 | June | January | 4.9 |
| Irminger Sea | $-0.73 \pm 1.13$ | 1626 | August | March | 0.6 |
| Greenland Sea | $-1.12 \pm 0.15$ | 729 | June | February | 7.5 |
| Norwegian Sea | $-0.25 \pm 1.08$ | 466 | September | January | 0.2 |
| Denmark Strait | $-2.24 \pm 0.28$ | 729 | February | August | 8 |
| Iceland-Scotland | $-2.27 \pm 0.72$ | 1139 | July | January | 3.1 |
| Newfoundland | $-3.71 \pm 1.85$ | 2125 | June | March | 2 |
| Rockall | $-1.05 \pm 0.59$ | 1379 | August | February | 1.8 |

The magnitude of the seasonal variability of $W_\Sigma$ is exhibited in Fig. 8, where the time series of $W_\Sigma$ at $z_{\min}$ with (solid
lines) and without (dashed lines) the seasonal signal are depicted for each region (Fig. 7 and Table 2). It displays a negative
$W_\Sigma$ all year round for the Labrador Sea that varies between $-2$ Sv (winter) and $-5$ Sv (late spring/summer). This result for
the Labrador Sea qualitatively agrees with Holte and Straneo (2017), who also found the strongest sinking in spring ($-1.2$
Sv) and the weakest sinking in winter ($-0.6$ Sv). Georgiou et al. (2019) also found this intensification of the sinking during
spring in their idealized Labrador Sea model in response to the larger density gradients between the boundary and the interior.
The Irminger and Norwegian Seas share a large temporal variability that is reflected in an almost identical standard deviation
of $\sim 1.1$ Sv (Fig. 7B,D), and a seasonal variability of $\sim 3$ Sv, similar to that found in the Labrador Sea (Fig. 8B,D). For
the Irminger Sea $z_{\min}$ remains constant during the year while for the Norwegian Sea it changes every season with an abrupt
deepening in winter when it reaches a depth of $\sim 1200$ m (horizontal dashed black lines in Fig. 8B,D). Contrarily, the Greenland
Sea shows a weak seasonal variability of $W_\Sigma$ at $z_{\min}$ (0.5 Sv, Fig. 8C), but the depth of largest sinking shallows significantly
during winter to $\sim 100$ m (black dashed lines in Fig. 7C). In terms of SNR, the Labrador Sea has a high value of $\sim 5$, although
smaller the Greenland Sea (7.5) where the seasonal signal is hardly identifiable. On the contrary, the Irminger and Norwegian
Seas yield low values of SNR (0.6 and 0.2 respectively, Table 2), which reflect their remarkable seasonal variability (Fig.
8B,D). The boundary sinking is delayed from the occurrence of deep convection in the interior of subpolar North Atlantic as
demonstrated by the fact that the largest boundary sinking in Labrador and Irminger Seas occurs in late spring/summer while
the deep convection takes place in late winter/early spring (Fig. S2).



**Figure 7.** Vertical profile of annual mean (thicker line) and monthly averages (thinner lines) of net vertical transport ($W_\Sigma$) for the regions defined in Fig. 6. $\mu$ and $\sigma$ are the climatological mean and standard deviation of $W_\Sigma$ at the depth of largest sinking (or minimum $W_\Sigma$, see legend). $z_{\min}$ is the depth where the largest sinking is found. Max and Min refer to the months with maximum and minimum mean $W_\Sigma$ (smallest and largest sinking) respectively. Seasonal mean depths of $W_\Sigma$ are displayed by horizontal black dashed lines.



Next, we evaluate to what extent the sinking regimes proposed in Sect. 3.2 for the entire subpolar North Atlantic are also
applicable to the individual regions of interest. To this end we have plotted the accumulated $W_\Sigma$ from the coast to the interior
at the depth of largest sinking for each region (Fig. 9). Overall, the Labrador, Greenland and Norwegian Seas show the sinking
regimes proposed in the Sect. 3.2, with a stronger negative accumulated $W_\Sigma$ near the slope at distances shorter than 200 km and
a small accumulated $W_\Sigma$ at larger distances. In particular, the Labrador Sea yields an accumulated sinking $W_\Sigma$ of around $-4$
Sv in the region covered by the sinking regime **I**, which is larger in magnitude than the $-1.5$ Sv or the $-0.5$ Sv obtained for the
Norwegian and Greenland Seas respectively for the same region (Fig. 9A,C-D). As for the entire subpolar North Atlantic, the
amount of negative accumulated $W_\Sigma$ in the regime **II** is generally smaller in magnitude: roughly $-1.5$ Sv for the Labrador Sea
and $-0.5$ Sv for Greenland and Norwegian Seas (Fig. 9A,C-D). Differences between sinking regimes **I** and **II** are subtle but
still distinguishable as the slightly larger seasonal variability (up to 2 Sv for the Labrador and Norwegian Seas) and the higher
number of oscillations at distances within the regime **II**. In contrast, the Irminger Sea shows a succession of positive/negative
accumulated $W_\Sigma$ over the distances covered by the regimes **I-II**, with a negative accumulated $W_\Sigma$ at 290 km of around $-1$ Sv.
Depending on the marginal sea considered regime **III** is found from 250-300 km off the coast, and it is associated with larger
monthly variations of accumulated $W_\Sigma$ than in regimes **I** and **II**. Some clear examples of this seasonality are exhibited by the
Irminger, Norwegian and Labrador Seas with ranges $\sim 3$ Sv (Fig. 9A-B,D); the Greenland Sea shows a weaker seasonality
(Fig. 9C) in line with Fig. 8C. We note that the pattern of accumulated $W_\Sigma$ is particularly complex for the Irminger Sea on
and near the slope, with positive accumulated $W_\Sigma$ at a distance to the coast between 100 and 170 km followed by negative
accumulated $W_\Sigma$ between 200 and 300 km off the coast. One explanation for that succession of upward/downward moving
waters near the boundaries is the uphill/downhill flow along the coast, which is probably linked to the bathymetry (as seen in
Fig. 2A, and also supported by a depth of largest sinking that does not vary seasonally in Fig. 7). At distances of more than
350 km from the coast, the Norwegian and Irminger Seas show a positive accumulated $W_\Sigma$ during winter yielding a seasonal
variation with respect to summer months of $2.5 - 3$ Sv. To summarize, our results confirm that to a large extent the proposed
sinking regimes remain applicable to marginal seas, in particular to the Labrador Sea. However, the mentioned differences and
similarities among the marginal seas (e.g. their different $z_{\min}$) reveal a complex picture, where the boundaries between the
different regimes are not fixed and can shift a few tens of km closer to or farther from the coast. Thus, sinking regimes are
influenced by the bathymetric features and local processes in each marginal sea.
**4.2   Overflow regions**
The Denmark Strait and the Iceland-Scotland Ridge are regions where the $W_\Sigma$ is mainly dominated by the overflow of waters
from the Nordic Seas towards the northern subpolar North Atlantic. Fig. 7E-F shows that the mean magnitude of $W_\Sigma$ is very
similar in both regions and amounts to roughly $-2.2$ Sv. Altogether it gives a total value of $\sim -4.5$ Sv for overflow waters,
which represents at most 33% of the total $W_\Sigma$ at $z_{\min}$ (compare overflow regions against Domain in Table 2). The outcome
from the Denmark Strait is in agreement with the $-2.2$ Sv estimated by Katsman et al. (2018) for the ORCA025 hindcast
(note that they estimated $W_\Sigma$ in a different area and $z_{\min}$) and 0.25 Sv higher than the transport found by Köhl et al. (2007)
in a model simulation. However, these model-based results are about 1 Sv weaker than the hourly observations, which yield



$-3.2\pm1.5$ Sv (Jochumsen et al., 2017). $z_{\min}$ differs for both regions, and is shallower for the Denmark Strait (729 m) than for
Iceland-Scotland (1139 m). This is related to the respective sill depths in the model.

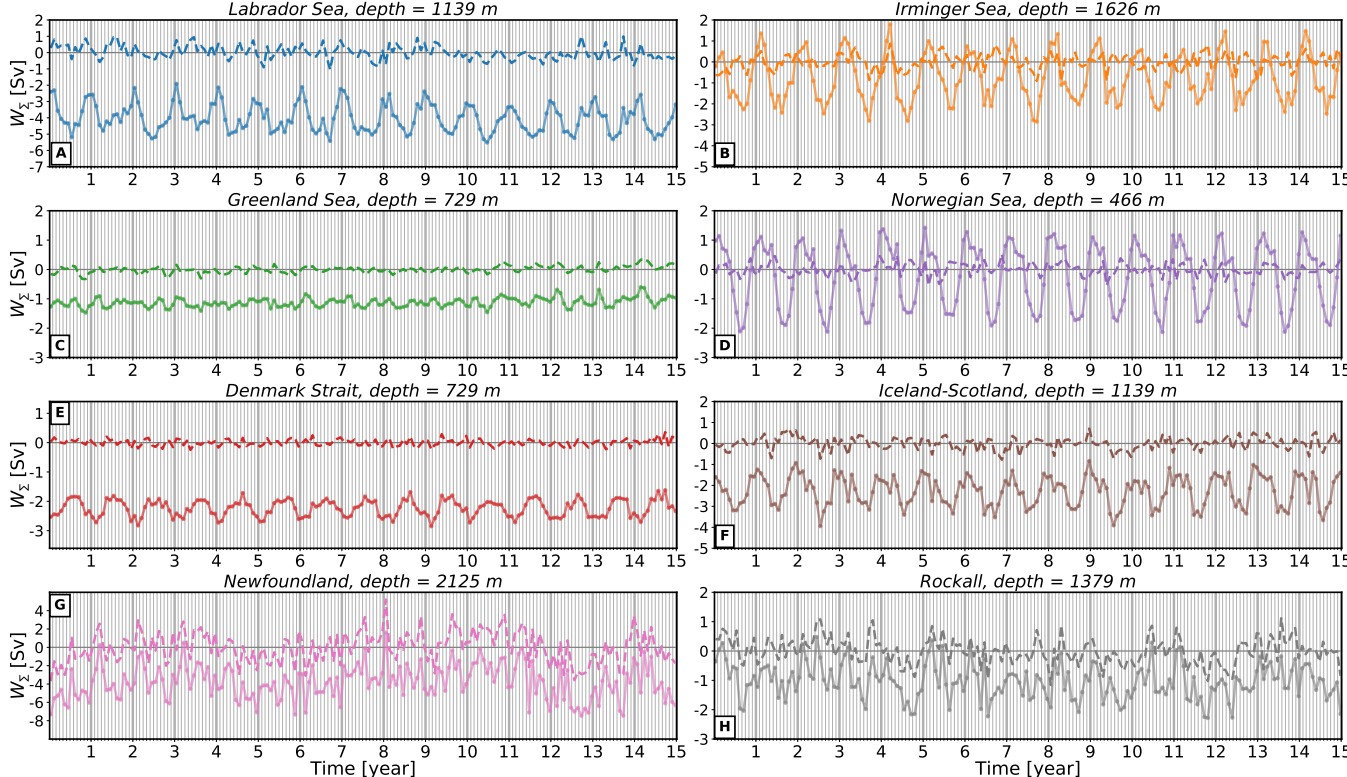

**Figure 8.** Time series of net vertical transport ($W_\Sigma$) at the depth of minimum $W_\Sigma$ ($z_{\min}$). This depth varies for each region according to Table 2 (see the plot title). The solid line includes the seasonal cycle; the dashed line shows the deseasoned time series. The seasonal cycle has been subtracted by removing the corresponding climatological monthly mean from every month. Note the different vertical scale of the various plots.

The downward transport in the Denmark Strait peaks in February and is weakest in July, which is out of phase with all other
regions; the Iceland-Scotland region peaks in August and is weakest in January (Table 2). Seasonal variability is present in
both time series with ranges of 0.6 Sv for the Denmark Strait and 2 Sv for Iceland-Scotland at their respective $z_{\min}$ (solid lines
in Fig. 8E-F, and Table 2). Besides this depth shows larger seasonal variations in Iceland-Scotland than in the Denmark Strait
(black horizontal dashed lines in Fig. 7E-F). The seasonal signal is smaller in the Denmark Strait than in other basins, with
differences between winter and summer (Fig. 8E) likely due to fluctuations of the overflow plume (Jochumsen et al., 2017;
Håvik et al., 2017), which has an observed reduced transport in summer. As other high-resolution models, this model simulation
tends to overestimate seasonal changes of overflow waters through the Denmark Strait: observations indicate a seasonal signal
of only around 0.05 Sv (Jochumsen et al., 2012). Moreover, sinking in Iceland-Scotland displays larger fluctuations than in the

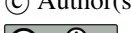



1  Denmark Strait ($\mathrm{SNR} = 3.1$ against $\mathrm{SNR} = 8$). This can be explained by its larger extent, covering not only the ridge itself but

2  also the surroundings, where both eddy-induced and near-boundary sinking may exist.

**Figure 9.** Accumulated net vertical transport ($W_\Sigma$) with respect to the distance to the closest land point. Distances are based on Fig. 4 (inset map). Annual (dashed black line) and monthly mean (colored lines) curves are depicted for the regions defined in Fig. 6. The accumulated $W_\Sigma$ has been calculated at the depth of minimum time-mean $W_\Sigma$ ($z_{\min}$), which differs for each region (see Table 2 and plot title). The bounds separating the sinking regimes (**I-II-III**) proposed in Fig. 4 (110 km, 290 km) are indicated with thicker solid vertical lines. Note the differences in the horizontal and vertical scales in the plots.



More differences between the Denmark Strait and the Iceland-Scotland regions are obvious from Fig. 9E-F where the time-
mean accumulated $W_\Sigma$ is plotted versus the distance to the coast. The positive and negative accumulated $W_\Sigma$ in the Denmark
Strait over the first 250 km off the coast (Fig. 9E) reflect waters moving southward from the Nordic Seas that first flow up and
then down over the sill. This is illustrated by the deepening of the isopycnal in Fig. S5C after crossing the Denmark Strait. In
Iceland-Scotland the sinking regimes **I** and **III** can be identified for distances to the coast smaller than 100 km and between
250 and 500 km respectively —note the two sill steps that appear in Fig. 9F: one near Iceland at $\sim 80$ km, and the other near
Scotland at $\sim 200$ km—. The distinction between regimes **I** and **II** is hardly applicable to the Denmark Strait due to the major
importance of the overflow waters. So our results suggest that the sinking regimes do not capture some specific features of the
overflows since they are governed by a different dynamics. Indeed, the location where sinking associated with overflows occurs
is not determined by lateral boundaries but rather by the bathymetry, so that it can occur at distances to the coast distinct from
the pattern shown by the marginal seas.

## 12     4.3    Mid-latitude seas

Newfoundland is located further south, in the vicinity of the Gulf Stream, while Rockall occupies the east Atlantic between
$5^\circ$ W and $25^\circ$ W. Together they yield a time-mean $W_\Sigma$ contribution of $\sim -4.8$ Sv at $z_{\min}$ (Fig. 7G-H). Newfoundland is the
region with the second largest $W_\Sigma$ after the Labrador Sea, with a sinking of $-3.7$ Sv. A significant difference between both
regions is $z_{\min}$, which is much deeper in Newfoundland (2125 m) than in Rockall (1379 m). Indeed sinking extends deeper
in Newfoundland, even reaching depths below 3000 m (Fig. 7G). The minimum (maximum) $W_\Sigma$ at $z_{\min}$ occurs in summer
(winter) for both areas, being in June (March) and August (February) for Newfoundland and Rockall respectively. Although
monthly variations reach 4 Sv for Newfoundland and 2 Sv for Rockall (solid curves in Fig. 8G-H), the seasonal cycle is not
very pronounced for either of the two regions at $z_{\min}$. The much larger temporal variability for Newfoundland is reflected in
$\sigma = 1.85$ Sv against the $\sigma = 0.59$ Sv of Rockall, despite the SNR is almost the same (2 against 1.8).
Clear differences are seen when we compare the accumulated $W_\Sigma$ with respect to the distance to the coast (Fig. 9G-H
for the two regions). Interestingly, Newfoundland displays large oscillations of positive and negative accumulated $W_\Sigma$ with
wavelengths of about 200 km. This suggests the presence of permanent mesoscale eddies in the interior, which are able to
induce those such strong vertical velocities, as shown in Fig. 5. In contrast, Rockall shows a quasi-linear decrease of the mean
accumulated $W_\Sigma$ with respect to the distance to the coast. It also displays seasonal variability that, for example, yields a smaller
sinking during late winter and spring months. We conclude that mean features of sinking in Newfoundland and Rockall regions
show similar characteristics to those seen for the entire subpolar North Atlantic, as reflected by some boundary sinking in
Rockall (regime **I**) and the strong eddy-induced velocities in the interior of Newfoundland (regime **III**).

## 30     4.4    Further characterization of sinking regimes illustrated by selected regions

In this Section we discuss in more detail the differences in sinking based on three regions: the Labrador and Irminger Seas,
and Newfoundland. These regions represent around 2/3 of the total sinking, are relatively far from overflows (although some
contribution in the northern Irminger Sea may be expected) and present remarkable differences in their dominant sinking





regimes covering a wide range of patterns that can be extended to other marginal seas. The spatial distribution of time-mean
vertical transport ($W$) for the three regions at the corresponding depth of largest sinking ($z_{min}$), which differs for each region
according to Table 2, is shown in Fig. 10A; the difference between the $W$ calculated during the months of minimum and
maximum $W_\Sigma$ is shown in Fig. 10B (also these months, distinct for each region, are indicated in Table 2). We have added black
contours to illustrate the positive/negative variation of the climatological mean EKE between the respective months at $z_{min}$
(Fig. 10B).

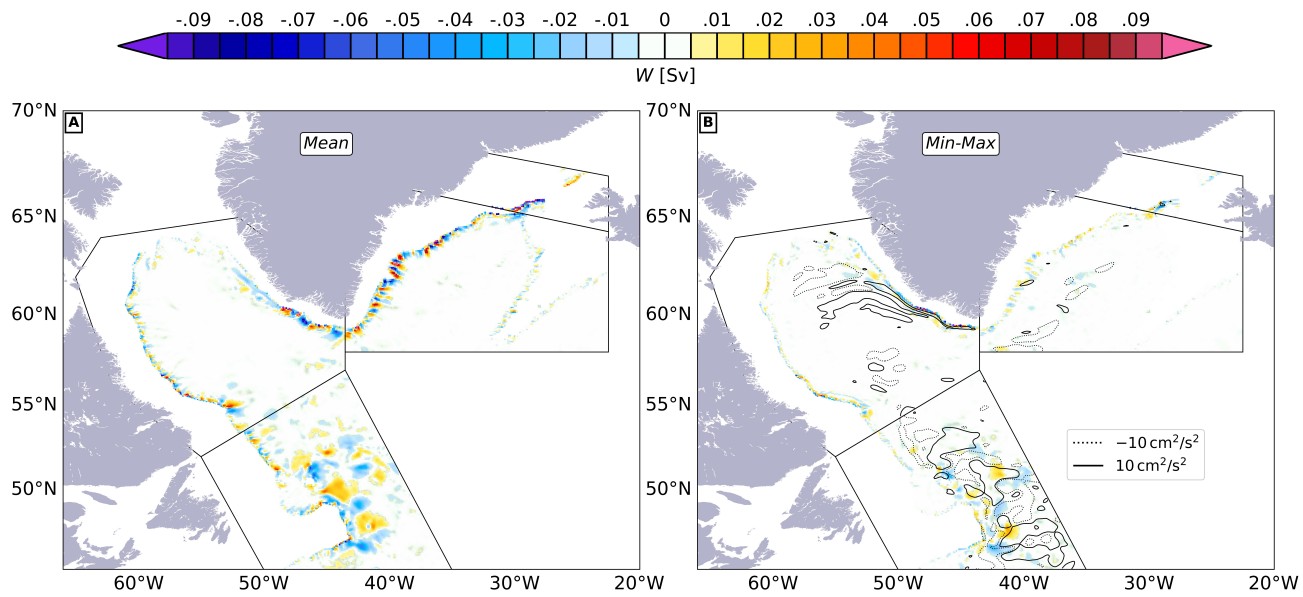

**Figure 10.** (A) Composite map of mean vertical transport ($W$) for the western regions of study (defined in Fig. 6) at the corresponding depth of minimum time-mean $W_\Sigma$, which differs for each region according to Table 2. (B) Same as (A) but now the mean $W$ (shading) and EKE (blacks contours, see legend) at the month of minimum $W_\Sigma$ minus the mean $W$ at the month of maximum $W_\Sigma$ is plotted. These months of minimum and maximum sinking also change for each region according to Table 2. Units of $W$ in Sv and of EKE in $\mathrm{cm}^2 \cdot \mathrm{s}^{-2}$.

The Labrador Sea, which yields the largest contribution to the time-mean $W_\Sigma$, is the most representative sea where all the
necessary ingredients for boundary sinking are fulfilled: a cyclonic boundary current, a steep bathymetric slope, an active eddy
field, and strong along-shore and cross-shore density gradients (Brüggemann et al., 2019). As a result, the three sinking regimes
proposed for the entire subpolar North Atlantic in Fig. 4 were also identified in Fig. 10A for the Labrador Sea. The strong $W$
near the boundary in the Labrador Sea (Fig. 10A) intensifies at the western side of the southern tip of Greenland during late
spring, which is associated with a nearby increase of EKE (see the solid black contours over the blue patches in Fig. 10B,
and the patches of EKE in Fig. S4C-E around the tip). Indeed the Labrador Sea displays an increase in the average horizontal
speed of the boundary current at $z_{min}$ during spring (Fig. 11A), which may enhance the horizontal velocity shear. As a result,
we can expect an increase in the mean advection of vorticity between the coast and near the peak of the boundary current (at
around 90 km, mostly within the region covered by regime **I**, Fig. 11A) and a decrease offshore of the location of the maximum





speed (mostly within the regime **II**) that may be compensated by the presence of more eddies. The more active role of eddies
exchanging waters between the interior and the boundary agrees with the reduced cross-shore potential density gradients found
in regions **I** and **II** (green and orange lines in Fig. 11B). The idea that changes in EKE pathways may facilitate to intensify
sinking is further discussed by Georgiou et al. (2019) for an idealized convective basin mimicking the Labrador Sea.

5       In contrast to the Labrador Sea, the change in $W$ in the Irminger Sea between the months of minimum and maximum time-

mean $W_\Sigma$ is significantly smaller (Fig. 10B). This finding, together with the permanent depth of largest sinking ($z_{\min}$) and
the up and down distribution of sinking found in Fig. 9B supports the hypothesis that the sinking near the boundary in the
Irminger Sea is mostly quasi-stationary and topographically-driven, which explains the small amount of net sinking found. In
Fig. 9B it can be observed that there is a large difference (in terms of seasonal variability) between the sinking regimes **I-II** and
the regime **III** in the Irminger Sea, which is probably driven by interior eddies. An interesting point to mention is the strong
mixing of waters within regions of regimes **I-II** (or equivalently, the reduction of the cross-shore gradient) during the months
of late winter and spring (orange and green lines in Fig. 11D). This pattern reflects a different behavior from what we find
in the Labrador Sea or Newfoundland (Fig. 11B,F) and is accompanied by an intensification of the boundary current during
the same months (orange and green lines in Fig. 11C). This difference is also reflected in the positive vertical transport during
winter within the regime **II** (blue and orange lines in Fig. 9B) and reinforces the crucial importance of topographic features in
driving the boundary sinking in the Irminger Sea.
Newfoundland yields the second largest contribution to sinking, which is largely produced within the sinking regime **III**. The
strong seasonal variations of $W$ in the Newfoundland region below 2000 m are mostly related to EKE interior pathway changes
(see black contours in Fig. 10B) impinged by North Atlantic Current fluctuations and meandering. This active contribution of
the large interior eddies in inducing these up and down strong vertical velocities has been already shown in Fig. 9G (also in Fig.
2A and Fig. 5), and is clear from Fig. 10A-B where a train of large eddies near the Flemish Cap is visible (also here the strong
EKE variations are denoted by the black contours). Finally, in the Newfoundland region the peak of the boundary current at its
corresponding $z_{\min}$ falls in regime **III** (Fig. 11E); although the largest to the boundary sinking occurs in the interior and below
2000 m, there is some sinking at shallower depths as indicated by its vertical structure in Fig. 7G. Similar plots showing the
overall weaker spatial patterns of sinking for the remainder of the regions can be found in Fig. S7 and Fig. S8.

## 5   Are regional variations in the net vertical transport connected to AMOC changes?

In Sect. 3 we demonstrated the consistency between the overturning streamfunction ($\psi_o$) at the southern boundary of our study
area at $45°$N and the total net vertical transport ($W_\Sigma$) in the subpolar Atlantic basin (Fig. 3B). As the amount of accumulated
negative $W_\Sigma$ appeared to differ between the boundaries and the interior, we have classified the sinking according to three
regimes (Fig. 4). Moreover, in Sect. 4 we have evaluated the spatial patterns and the seasonal variability of sinking at the
regional level. The fact that nearby areas exhibit striking differences in the amount, the seasonality and the distribution of $W_\Sigma$
suggests that, to a large extent, it depends on local dynamics and bathymetry. However, it still remains unclear how regional
boundary sinking is related to the AMOC. For instance, does a decrease/increase in Labrador Sea $W_\Sigma$ have any quantifiable





effect on the AMOC? To address this we have computed the cross-correlation between the reverted time series of the maximum
of $\psi_o$ at $45°N$ (red lines in Fig. 3B) and the time series of $W_\Sigma$ (Fig. 8) for each region at the depth of largest sinking ($z_{min}$,
Table 2). We have performed the analysis for two cases: with seasonal variability (Fig. 12B) and without seasonal variability
(Fig. 12C). Fig. 12A shows the monthly climatology of sinking at their corresponding $z_{min}$ so that the months with the largest
and smallest sinking for all regions (Table 2) are easily identifiable. Solid lines in Fig. 3B and in Fig. 9 include the seasonal
signal whereas in the dashed lines seasonality has been subtracted. A positive correlation at a positive time-lag $\tau$ means that
stronger (weaker) sinking yields a stronger (weaker) AMOC ($\psi_o$) at $45°N$ $\tau$ months later.

**Figure 11.** 15-year climatology of the following variables with respect to the distance to the coast (according to the inset map in Fig. 4) for (A)-(B) the Labrador Sea, (C)-(D) the Irminger Sea, and (E)-(F) Newfoundland: (A)-(C)-(E) horizontal speed of current at the depth of largest sinking ($z_{min}$, Table 2); (B)-(D)-(F) potential density anomalies ($\sigma_\rho = \rho - 1000$ [kg·m$^{-3}$]) averaged between $z$-layers 14 and 24 ($\sim 220 - 1650$ m). For all panels the dashed black line depicts the mean, whereas colored lines show the monthly average. The bounds between the sinking regimes proposed in Fig. 4 are indicated with thicker solid vertical lines. Note the differences in horizontal scales of the subpanels.





The high correlation between $W_\Sigma$ for the entire subpolar North Atlantic (DOMAIN) and $\psi_o$ ($> 0.9$, mentioned in Sect.
3) appears clearly at zero-lag for both study cases. Also the time-lags found for regions are in agreement with the temporal
separation between the corresponding months of minimum $W_\Sigma$ and the month of maximum $\psi_o$ at $45°$N. For instance, the
Labrador Sea displays the minimum sinking in June whereas $\psi_o$ has its maximum in August (Fig. 12A). As a consequence,
the highest correlation is found for a lag of the AMOC of 1-2 months. The same is found for all other regimes, including the
Denmark Strait, which has the minimum $W_\Sigma$ in February yielding a negative correlation at zero-lag. Fig. 12B shows that for
the southern regions (Rockall and Newfoundland) correlations are surprisingly weak. One reason for this is that the signal
of negative $W_\Sigma$ is very noisy for these two regions, with no clear seasonal cycle (Fig. 8G-H), while $\psi_o$ has a clear seasonal
signature. Also for the Greenland Sea correlations are rather weak ($< 0.4$), presumably due to its small seasonal variability
(Fig. 8C).
To eliminate the influence of seasonality we repeat the analysis on the deseasoned signal. Resulting correlations (Fig. 12C)
demonstrate that the only region with a significant correlation between variations of sinking and $\psi_o$ is Newfoundland. This
is the region with the largest non-seasonal variations (Fig. 8G, dashed line) and that shares its boundary with $\psi_o$ at $45°$N.
Therefore it is reasonable to think that any change in the North Atlantic Current, either in strength or position, will reflect in
sinking in Newfoundland and vice versa (for instance a fluctuation in the train of eddies nearby the Flemish Cap).
The existence of a high correlation does not necessarily implies that variations in the sinking waters govern the AMOC as
the different regions in this simulation are subject to the same strong large-scale forcing, for instance the seasonal heat fluxes or
wind stress variations that affect mid and high North Atlantic latitudes. Thus, the AMOC and the sinking are likely responding
synchronously to variations in large-scale forcing. Therefore our results using an Eulerian standpoint do not evidence that a
variation (marked increase/decrease) in $W_\Sigma$ at any of the marginal seas propagates to the lower cell of the AMOC. To investigate
this in more detail requires the use of a Lagrangian approach to track the boundary sinking and subsequent spreading of waters,
which is beyond the scope of this paper.





**Figure 12.** (A) Time series of the monthly climatology of $W_\Sigma$ for the regions defined in Fig. 6 at the depth of largest mean sinking ($z_{\min}$, which differs between regions according to Table 2). (B) Cross-correlation between the maximum of the reverted overturning streamfunction ($\psi_0$) at 45°N (red lines in Fig. 3B) and the regional time series of net vertical transport, $W_\Sigma$ (Fig. 8), at the depth of minimum time-mean sinking for a set of time-lags (in months). (C) As (B) but without the seasonal variability. The seasonality has been removed by subtracting the corresponding 15-year monthly mean (i.e. panel (A) for the regional time series). Sinking leads over $\psi_0$ for positive lags. DOMAIN refers to the whole study area (Fig. 6) and the acronyms are defined as: Labrador Sea (LS), Irminger Sea (IS), Greenland Sea (GS), Norwegian Sea (NS), Denmark Strait overflow (DSO), Iceland-Scotland Ridge overflow (ISO), Newfoundland (NF) and Rockall (RL), marginal seas (MS), overflow regions (OF) and mid-latitude seas (ML).





## 6  Summary and discussion

Based on a high resolution ocean model simulation forced by a prescribed annual cycle of wind, precipitation and heat fluxes, we have found that the amount of minimum time-mean net vertical transport ($W_\Sigma$) for the entire subpolar North Atlantic Ocean is consistent with the transport and vertical structure of the AMOC core at mid latitudes ($45^\circ$N), with an average of about $-14$ Sv at a depth of 1139 m. Moreover, the prescribed annual cycle introduces a strong seasonality that favours more sinking of waters at basin scale and a stronger AMOC during summer than in winter, with a similar seasonal variability in both signals ($\sim 10$ Sv). However, this picture becomes much more complex at regional scales, as is illustrated by the different depths at which the largest sinking occurs (ranging from 460 to 2000 m), the distinct spatial distribution and the asynchronous seasonal variations of $W_\Sigma$ that are found for the different regions in the subpolar North Atlantic.

In accordance with recent studies, our model results confirm that the largest vertical transports occur near the boundaries below the mixed layer depth (Katsman et al., 2018; Brüggemann et al., 2019; Georgiou et al., 2019), in a narrow band that extends between 50 and 300 km off the coastline. When we consider the sinking over the whole subpolar North Atlantic, three dominant sinking regimes are revealed: regime **I** ($\sim 0 - 100$ km) appears where the continental slope is steepest, regime **II** ($\sim 100 - 300$ km) covers the remainder of the continental slope, and regime **III** (distances $> 300$ km) occurs in the ocean interior. Our results indicate that around the $90\%$ of the accumulated sinking takes place in the area covered by regions **I** and **II**, while the largest seasonal variability of sinking occurs in region **III**. The near-boundary sinking in regimes **I** and **II** is thought to be governed by the ageostrophic dynamics discussed by e.g. Spall and Pickart (2001) and Straneo (2006), and its amount depends on the interplay of several factors: the existence of a boundary current on a pronounced topographic slope, the presence of eddies, and on the along-shore and cross-shore density gradients (sloping isopycnals) near the slope. This implies that the budget of $W_\Sigma$ is potentially sensitive to the intensity and the width of the boundary current, the strength of the eddy field, and the dominant eddy paths (Georgiou et al., 2019).

Distinguishing by regions, we find that most of sinking occurs in the Labrador Sea ($\sim -4.0$ Sv), Newfoundland ($\sim -3.7$ Sv) and the overflow regions (Denmark Strait and Iceland-Scotland Ridge with $\sim -2.2$ Sv each). The Irminger and Norwegian Seas show a strong seasonally-dependent behaviour, with sinking during part of the year and upwelling the rest of the year and hence little net sinking. We identified the three sinking regimes in almost all regions except in the overflow regions, which are governed by a different dynamics, and in the Irminger Sea. The Irminger Sea shows a distinct sinking dynamics near the boundary due to the existence of bathymetry-forced flows and probably by some overflow waters coming from the Denmark Strait. Moreover, in each region the distance from the coast that marks the boundary between the sinking regimes is seen to shift due to the local dynamics and bathymetric features (e.g. different steepness or shelf width) of each region.

The dominance of the seasonal forcing in the sinking response (probably induced by the repeated forcing conditions) limits us to find a connection between the regional variations of sinking and the lower cell of the AMOC at mid latitudes, though previous research has suggested a complex interaction between the surface atmospheric forcing, the boundary current and interior waters for which eddies play a crucial role (Georgiou et al., 2019). To gain insight into this connection would require an analysis of the near-boundary sinking and the AMOC in a model simulation with varying surface forcing. In addition, we



have studied the Eulerian net vertical transport without referring to the water-mass properties while the subpolar North Atlantic
is characterized by a strong densification of waters during late winter and spring. The latter would require an assessment of
sinking in density space, which is outside the main scope of this study, which focuses only on the vertical structure, seasonality
and spatial distribution of sinking. Also we note that monthly fields neither allow to quantify accurately which waters are
sinking, nor the amount of isopycnal and diapycnal mixing, since isopycnals significantly fluctuate at shorter time scales. In
this regard, our next step aims to address the above points by tracking the waters sinking near the boundary using a simulation
with higher temporal resolution. With this analysis, we expect to identify the characteristics of the near-boundary sinking water
masses, and to assess if any of their preferred pathways take them to the lower limb of the AMOC.
*Author contributions.*   JMS and CAK designed the paper, HAD provided the model data, JMS performed the data analysis, JMS and CAK
interpreted the results and wrote the paper. All authors have read and approved the submitted version of the manuscript.
*Competing interests.*   The authors declare that no competing interests are present.
*Acknowledgements.*   J.-M. Sayol and C. A. Katsman thank the financial support by the Netherlands Scientific Research foundation (NWO)
through the VIDI grant number 864.13.011 awarded to C.A. Katsman. Help in data processing from M. Kliphuis and comments from N.
Brüggemann, S. Georgiou, S. Ypma and C. van der Boog on the analysis and the manuscript are greatly appreciated.



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
