# Peer review of "Seasonal and regional variations of sinking in the subpolar North"

_Ocean Science, 2019_

## Referee Comment (RC1) · Anonymous Referee #1 · 25 Apr 2019

The present manuscript provides a useful synthesis of a number of recent studies using idealized models and observations of the vertical mass transport in buoyancy driven systems. While it does not present any fundamentally new ideas, it is a useful bridge between the idealized studies and the complexity of the actual North Atlantic system. It is well written and illustrated and I believe it could be published with few if any revisions. I make a few suggestions below that may help clarify a few points but will leave it to the editor to determine if they are ultimately necessary.

My first suggestion relates to the choice of origin for the offshore directed coordinate used in the analysis shown in Figs 4, 9, and 11, and the definitions of the Regimes

I- III. I suggest that a better choice for the origin would be the shelf break rather than the coast as the broad shelves (e.g. in the Rockall area) play no role in the dynamics under discussion.

A second, and somewhat related point, concerns discriminating between regions by the slope of the topography. At a number of points in the text (e.g the paragraph following Fig 7), the importance of "steep topography" in localizing vertical velocity is discussed. However, no direct quantitative relationship between topographic slope and vertical motion is provided.

Minor Points

Pg. 6, line 24 : Thea AMOC

Pg 6, line 31: A clarification that the maximum is taken over monthly means, rather than instantaneous values would be helpful here.

Pg 9, line 4: I am looking for a box or some other indication of the region in Fig 2. I assume you mean defined _by_ the domain shown in Fig 2?

Pg 15, Fig 8: I found little additional information in this Figure beyond that shown in Fig 7. and Table 2. Could be dropped without loss of support for the text.

---

## Referee Comment (RC2) · Anonymous Referee #2 · 10 May 2019

This manuscript analyzes the vertical velocity in the Atlantic north of 45N, using 15 years of a high resolution simulation forced by a repeating seasonal cycle. As shown in previous more idealized studies, there is net sinking in regions near the ocean boundary, which is highly correlated with the overturning streamfunction at 45N. The unique and original contribution of this manuscript is the examination of the spatial variability of the sinking. While the Labrador Sea is shown to provide a large fraction of the near-boundary downwelling, the Irminger sea has a more complicated signal, and the Newfoundland basin has a vertical motion largely driven by eddies in the interior. The temporal variability of the downwelling signals in these different regions is also different - the Labrador sea has a seasonal signal similar to that of the basin-wide overturning,

while the Newfoundland basin has non-seasonal temporal variability driven by eddies. Overflow regions are also shown to behave differently. This paper therefore provides an insight into where and how the downwelling necessary for the AMOC occurs. I therefore support publication. I do however have a few concerns and suggestions for the authors, requiring revision before acceptance.

General concerns

1. The overflows The authors indicate that while the overflow regions do contain significant and important downwelling (together producing as much downwelling as the Labrador sea), the overflow regions do not show the same downwelling mechanism controlled by the lateral boundaries as, for example the Labrador Sea. The authors briefly mention that a key distinction is that the downwelling in the overflow regions is more controlled by bathymetry. I would like to see more attempt to examine the mechanisms in the overflows. Rather than show downwelling aggregated by distance from the nearest land, what would be a better aggregation in the overflow regions - perhaps distance downstream from the sill? I also wonder if the boxes shown for the overflow regions are the best choices, since the downwelling dynamics downstream of the sills could be quite different from upstream. What is the rationale for the ISO box which extends far upstream into the GIN sea? If the DSO box extended further into the Irminger sea (e.g. as far as the region of traceable descent of the dense water downstream of the sill) perhaps the signal from the smaller remaining Irminger box would be less "contaminated"? In summary, I encourage more specific investigation of the overflow regions, using aggregation more suitable to the dynamics in these regions.

2. The overturning circulation The overturning circulation shown in figure 1A has several features which may not be realistic - e.g. the large decrease in stream function north of 35N, the very shallow depth, particularly north of 45N. See Dunne et al, 2012 JoClimate, figure 4 for examples of overturning which extend further north and deeper. What connection might there be between the diagnosed pattern of overturning, and the dominant horizontal and vertical locations of downwelling seen in this simulation?

3. Choice of aggregation by distance from the coast. For the overflows in particular, but also for some other regions (e.g. wide shelves compared to narrow shelves), I wonder if there might be a better method of aggregating the downwelling than by distance from the coast. For example, would bathymetric depth be more physically meaningful than distance from the coast? Figure 4 inset shows that distance from the coast gives undue influence to Jan Mayen island in the Greenland sea, for example. If the downwelling occurs next to the topography, we might expect to see a close relationship between the depth of maximum downwelling and the bathymetric depth, which is hidden when aggregating by distance from land.

4. Interpretation of upwelling I find it hard to interpret the upper ocean/near boundary upwelling seen in the Irminger sea (figure 7B and 9B), and the near boundary up-welling seen in the Denmark Straits (figure 9E). What does this imply physically - how is the circulation closed, where is the upwelled water going? Would a section, showing velocities binned by horizontal and vertical distance, help provide more information?

5. Small scale motion of alternating sign In figure 2A, I have difficulty seeing a clear signal of net downwelling in any region around the northern boundary because of the large amplitude signals of alternating sign. Whereas the mean Labrador sea signal in figure 7 is for near-boundary downwelling, the mean Irminger sea signal is for near-boundary upwelling, yet in figure 2A, the two regions look qualitatively similar, with large amplitude alternating signals. Is it possible to apply a spatial filter to the fields in figure 2A, to show what the vertical motion looks like when averaged over the eddy scales? It's also not clear to me why the Irminger sea alternating signal is suggested to be tied to topography - are there features in the bathymetry on these scales? If so, aggregating by bottom depth might be a way to clarify.

Specific comments

Page 2, Line 3: Insert "few" after "Over the last"

Page 2, Line 31: Change "looses" to "loses"

Page 3, Line 28: Change "and idealized eddy-resolving model" to "an idealized eddy-resolving model".

Page 4, line 3-4: "not adding even more complexity" - I'm not sure what you mean by this, perhaps not adding the additional complexity of historical variations in surface forcing?

Page 6, line 20-21: Change "Nevertheless it can be inferred a stronger transport in winter than in summer" to "Nevertheless, a stronger transport in winter than in summer can be inferred..."

Figure 2B: The depth color bar shows increasing depth upward, whereas it is more intuitive to show it downward on the color scale (a minor point, I know).

Figure 3: Given this downwelling in Sv, if you divided through by the area at each depth, what would this imply for the magnitude of the vertical velocity in m/s? How does this compare with the magnitudes shown in figure 2A? (I'm guessing that the noise shown in figure 2A is much greater than the mean signal).

page 10, line 4-5: If the W value is shown at a depth of 1139m, what do you do if the grid cell bathymetry is shallower than that value?

p11 discussion of figure 5, and figure 5: To see the context of the currents around 1250km-1000km south of Greenland, it would be helpful to show an inset of the surface currents over the whole North Atlantic. How is this "standing eddy" related to the North Atlantic Current?

page 26, lines 30-31: Change "limit us to find" to "prevents us from finding"

page 27, line 5: Change "isopycnals significantly fluctuate" to "isopycnals fluctuate significantly"

---

## Short Comment (SC1) · 17 May 2019

Dear authors,

I have a question regarding the particular region of Flemish Cap, where you report a relatively important eddy-driven sinking.

I wonder how the latter reconciles with the key ingredient for vertical sinking: a positive along-stream density gradient supporting an in-shore geostrophic flow in the upper layer. Through mass conservation this drives downwelling along the slope, as the authors well explained in their introduction.

[Figure]

This is not the case of the boundary current encircling Flemish Cap, which is warming as it flows downstream due to exchanges with the warmer North Atlantic Current, so that the sign of the along-stream density gradient would rather suggest upwelling there.

Could the authors comment on this ?

Many thanks in advance.

A very interested reader.

——————————————————

---

## Author Comment (AC1) · 25 Jun 2019

Please, find in the attached file our reply to your comments/suggestions (Pages 3-7). The manuscript with tracked changes is also attached after page 22.

Please also note the supplement to this comment:
https://www.ocean-sci-discuss.net/os-2019-27/os-2019-27-AC1-supplement.pdf

———————————————

---

## Author Comment (AC2) · 25 Jun 2019

Please, find in the attached file our Reply to your comments/suggestions (in Pages 8-18). See also the new version of the Manuscript with tracked changes (after page 22).

Please also note the supplement to this comment:
https://www.ocean-sci-discuss.net/os-2019-27/os-2019-27-AC2-supplement.pdf

———————————————

---

## Author Comment (AC3) · 25 Jun 2019

Please, find attached our Reply to your comment (Pages 19-21). You can also find the new version of the Manuscript with tracked changes after page 22.

Please also note the supplement to this comment:
https://www.ocean-sci-discuss.net/os-2019-27/os-2019-27-AC3-supplement.pdf

————————————————

---

## Author Response (AR2)

Dr. Matthew Hecht
Handling Topic Editor
Ocean Science

Dear Editor,

Please find attached a new version of the manuscript with number **os-2019-27**, "Seasonal and regional variations of sinking in the subpolar North Atlantic from a high resolution model" which has been modified according to the suggestions and criticisms of two Referees (RC1 and RC2). We also attach a detailed reply letter to the Referees with a list of changes included in the new version of the manuscript, and a version with tracked changes. For completeness, we have included the answer to a short comment (SC1) from Dr. Damien Desbruyères.

We thank both Reviewers to their comments, which have helped us to improve the manuscript and to clarify some fuzzy points. We also thank their appreciation of our work and their generally positive comments. We note that, after our revision, the key conclusions of this work remain unaltered.

The main changes to the manuscript are:

– As suggested by Referee 2, in order to better isolate the overflows in the Iceland-Scotland Ridge, the boundaries of the eastern regions in which we have divided the subpolar North Atlantic have been modified. Numbers in Table 2 and Fig. 6-12 have been updated accordingly. The description of these figures in Sect. 4 has been also updated with the new values for the sinking.

– According to a comment from Referee 2, a new figure (Fig. 9) has been added to the main manuscript describing the vertical transport in the Denmark Strait with an alternative reference system (vertical transport accumulated from the sill).

– Two complementary figures have been added to the supplementary material. The first one, motivated by comments from Referee 2, helps to clarify the up/down bathymetric forced movement of waters in the Irminger Sea (Fig. S9). The other, inspired by the short comment by Dr. Desbruyères, shows more clearly the strong vertical velocities at the edge of the large eddies that develop from the North Atlantic Current in the interior of Newfoundland, nearby the Flemish Cap (Fig.

S4).

– As suggested by Referee 1, Fig. 8 has been moved to the supplementary material (now Fig. S8).

– The origin of the calculation of the accumulated sinking has been shifted from the coastline to depth contour of 50 m. This clarifies the figures, while the conclusions on the existence of different sinking regimes remains unchanged. Fig. 4-5-8 have been updated accordingly.

Other minor modifications are clearly stated in the attached Reply.

We expect that, with the changes and clarifications we have made, our paper will be found suitable for publication in Ocean Science.

Yours sincerely,
Dr. Juan-Manuel Sayol, on behalf of all authors

**Referee 1 (RC1):**

**The present manuscript provides a useful synthesis of a number of recent studies using idealized models and observations of the vertical mass transport in buoyancy driven systems. While it does not present any fundamentally new ideas, it is a useful bridge between the idealized studies and the complexity of the actual North Atlantic system. It is well written and illustrated and I believe it could be published with few if any revisions. I make a few suggestions below that may help clarify a few points but will leave it to the editor to determine if they are ultimately necessary.**

■ **Reply 0.** We thank the Referee for his/her comments and time employed in carefully reviewing our work. A point-by-point reply to all questions/concerns/suggestions can be found below. ————————————————————————————————————

**Major points**

**1. My first suggestion relates to the choice of origin for the offshore directed coordinate used in the analysis shown in Figs 4, 9, and 11, and the definitions of the Regimes I-III. I suggest that a better choice for the origin would be the shelf break rather than the coast as the broad shelves (e.g. in the Rockall area) play no role in the dynamics under discussion.**

■ **Reply 1.** Thank you for this comment, which is also raised by Referee 2 comment 3. The choice of the coast as the origin to accumulate vertical transport has allowed us to uncover sinking regimes (**I-III**), which we believe are useful to reveal qualitatively at which regions the boundary sinking tends to dominate and at which not. However, we agree that the distance to the coast is not very suitable for some regions, such as the overflow areas or Newfoundland. In order to remove part of the undesired shallower shelf from the plots at which no boundary sinking is produced -at the depth of largest sinking-, we have shifted the origin from the coastline to the bathymetric contour of 50 m ($C_{50}$). So now vertical transport accumulates starting at the closest grid point to a bottom depth of 50 m, and later accumulates other grid points located farther according to their distance to $C_{50}$. The clearest positive effect is seen in the Labrador Sea, where now sinking starts at 10-20 km from $C_{50}$ in turn of the former 50 km (see Fig. R1 below and Fig. 8 in the new version of the manuscript—note that the eastern regions have been redefined in order to better isolate the Iceland-Scotland overflows as requested by Referee 2, see Fig. R2 or Fig. 6 in the manuscript). Note that according to this new origin, we have redefined the position of the boundaries of sinking regimes, now at 90 and 250 km rather than 110 km and 290 km (Fig. 4 in the manuscript). However, qualitatively results remain very similar, especially for the case of the overflows. For this reason and to give another view (also requested in the major point 1 from Referee 2) an extra Figure (Fig. 9 in the new version of the manuscript) has been added, at which sinking in the Denmark Strait is accumulated according to the distance from the sill that separates the upward and downward movement of water masses. ————————————————————————————————————

**2. A second, and somewhat related point, concerns discriminating between regions**

[Figure]

Figure R1: Accumulated net vertical transport ($W_\Sigma$) with respect to the distance to the closest bathymetric contour of 50 m ($C_{50}$). Distances are shown in Fig. 4 (inset map) of the manuscript. Annual (dashed black line) and monthly mean (colored lines) curves are depicted for the regions defined in Fig. 6. The accumulated $W_\Sigma$ has been calculated at the depth of minimum time-mean $W_\Sigma$ ($z_{min}$), which differs for each region (see Table 2 and plot title). The bounds separating the sinking regimes (**I-II-III**) proposed in Fig. 4 of the manuscript (90 km, 250 km) are indicated with thicker solid vertical lines. Note the differences in the horizontal and vertical scales in the plots.

**by the slope of the topography. At a number of points in the text (e.g the paragraph following Fig 7), the importance of "steep topography in localizing vertical velocity is discussed. However, no direct quantitative relationship between topographic slope and vertical motion is provided.**

■ **Reply 2.** Thank you for this comment, it is an opportunity to clarify it. Steep topography is one of the ingredients that contribute to boundary sinking together with the progressive densification of the boundary current in alongshore direction, a crosshore gradient of density between the interior and the shelf and the existence of eddies. However, a steep topography by itself is not enough to induce boundary sinking since densification of the boundary current is necessary (Spall, 2010; Brüggemann et al., 2017; Brüggemann and Katsman, 2019). One way to isolate the relevance of topography in the boundary sinking is shown in Fig. R3, in which for the whole basin and for each region the depth of largest sinking is plotted against the bottom depth of the corresponding grid point. Overall, it shows that the strongest sinking tends to occur rather close to the bottom, highlighting the relevance of topography. A remarkable exception is Newfoundland, where there is a cloud of points on the right side rather misaligned, likely reflecting the contribution from interior eddies. We have not added Fig. R3 to the manuscript nor in the supplementary material since a similar information can be deduced from Fig. 2B, in which the depth of strongest sinking is very close to the bathymetric contours. Also the new Fig. S9 provides information in the same direction in the Labrador and Irminger Seas. We have reviewed carefully all our expressions in the manuscript that can lead to any misinterpretation regarding the role of steep topography. For instance in the conclusions (Page 28, Ln 16-19):

*"The near-boundary sinking in regimes **I** and **II** is thought to be governed by the ageostrophic dynamics discussed by e.g. Spall (2010) and Straneo (2006), and its amount depends on the interplay of several factors: the existence of a boundary current, a steep slope, the presence of eddies, and on the along-shore and cross-shore density gradients (sloping isopycnals)."*
* * *
**Minor points**

**1. Pg. 6, line 24 : Thea AMOC.**

■ **Reply 3.** Corrected. ─────────────────────────────────

**2. Pg 6, line 31: A clarification that the maximum is taken over monthly means, rather than instantaneous values would be helpful here.**

■ **Reply 4.** Thanks. Now it reads (Page 7. Ln 2):

*". . . obtained from the monthly mean fields"*
* * *
**3. Pg 9, line 4: I am looking for a box or some other indication of the region in Fig 2. I assume you mean defined _by_ the domain shown in Fig 2?**

[Figure]

Figure R2: Map of the North Atlantic $[66\,\mathrm{W} - 20\,\mathrm{E},\ 45\,\mathrm{N} - 75\,\mathrm{N}]$ divided into eight regions. DSO and ISO refer to Denmark Strait and Iceland-Scotland Ridge overflow regions respectively. The surface area of each region is shown in the legend in $10^6 \cdot \mathrm{km}^2$.

■ **Reply 5.** Sorry for the lack of clarity. As suggested, it has now been explicitly stated in Page 8. Ln 10.:

*"Second, we sum $W$ over the horizontal domain shown in Fig. 2."*

**4. Pg 15, Fig 8: I found little additional information in this Figure beyond that shown in Fig. 7 and Table 2. Could be dropped without loss of support for the text.**

■ **Reply 6.** We agree that is not a key figure to understand the main points of this work. However, it nicely shows the seasonal cycle differences among regions, being referred few times in the manuscript. Accordingly this figure has been moved to the supplementary material and now is referred in Sect. 4 as Fig. S8.

[Figure]

Figure R3: Scatter plots showing the bottom depth (h) and the depth of the mean minimum vertical transport ($\overline{W}_{min}$) for the whole domain (the complete area of Fig. R2) and for each of the regions shown in Fig. R2. Only those grid points with $\overline{W} < 0.02$ Sv are shown. The black line indicates those points where the depth of largest sinking matches the bottom depth.

**Referee 2 (RC2):**

This manuscript analyzes the vertical velocity in the Atlantic north of $45°$N, using 15years of a high resolution simulation forced by a repeating seasonal cycle. As shown in previous more idealized studies, there is net sinking in regions near the ocean boundary, which is highly correlated with the overturning streamfunction at 45N. The unique and original contribution of this manuscript is the examination of the spatial variability of the sinking. While the Labrador Sea is shown to provide a large fraction of the near-boundary downwelling, the Irminger sea has a more complicated signal, and the Newfoundland basin has a vertical motion largely driven by eddies in the interior. The temporal variability of the downwelling signals in these different regions is also different- the Labrador sea has a seasonal signal similar to that of the basin-wide overturning, while the Newfoundland basin has non-seasonal temporal variability driven by eddies. Overflow regions are also shown to behave differently. This paper therefore provides an insight into where and how the downwelling necessary for the AMOC occurs. I therefore support publication. I do however have a few concerns and suggestions for the authors, requiring revision before acceptance.

■ **Reply 0.** We thank the Referee for his/her comments and time employed in carefully reviewing our work. A point-by-point reply to all questions/concerns/suggestions can be found below. ─────────────────────────────────────────────

**Major points:**

1. The overflows: The authors indicate that while the overflow regions do contain significant and important downwelling (together producing as much downwelling as the Labrador sea), the overflow regions do not show the same downwelling mechanism controlled by the lateral boundaries as, for example the Labrador Sea. The authors briefly mention that a key distinction is that the downwelling in the overflow regions is more controlled by bathymetry. I would like to see more attempt to examine the mechanisms in the overflows. Rather than show downwelling aggregated by distance from the nearest land, what would be a better aggregation in the overflow regions - perhaps distance downstream from the sill? I also wonder if the boxes shown for the overflow regions are the best choices, since the downwelling dynamics downstream of the sills could be quite different from upstream. What is the rationale for the ISO box which ex-tends far upstream into the GIN sea? If the DSO box extended further into the Irminger sea (e.g. as far as the region of traceable descent of the dense water downstream of the sill) perhaps the signal from the smaller remaining Irminger box would be less "contaminated"? In summary, I encourage more specific investigation of the overflow regions, using aggregation more suitable to the dynamics in these regions.

■ **Reply 1.** We thank the reviewer for this comment. We agree that the surface of the region chosen to represent the Iceland-Scotland ridge (ISO) was too big, occupying part of the GIS seas. We have redefined all eastern Atlantic regions: Norwegian Sea, Greenland Sea, Iceland-Scotland Ridge and Rockall. A major consequence is the reduction of the sinking over the Iceland-Scotland Ridge region (ISO), where now only those areas corresponding to the two sills and surroundings have been included (see Fig. R2 of this reply or Fig. 6 in the revised version of the manuscript). Accordingly, Figures 6, 7, 8, 10, 11, 12, S8, S10, S11 and Table 2 have also been modified in the revised version of the manuscript. In terms of sinking, the main changes in the regions of the eastern subpolar Atlantic are (see Fig. 7 and Table 2 of the manuscript for a more detailed description of numbers):

- On average, the Norwegian Sea yields weak upwelling (0.37 Sv) at the depth of largest sinking, although with an increased variability reflected in rather intense net sinking (<-1 Sv) during summer months.

- The Iceland-Scotland region now presents the largest mean sinking at a shallower layer (at 918 m). This sinking is around 0.5 Sv larger than in the former version of the manuscript. The standard deviation is smaller, approaching a value closer to that in the Denmark Strait region.

- The Greenland Sea now reaches the largest sinking in the first 100 m, and it displays a larger seasonal variability. However, sinking remains strong down to a depth of 1500 m, with smaller seasonal variability below 500 m.

- The magnitude of sinking has increased by 0.5 Sv in Rockall. The reason is that the new region contains some sinking near southern Iceland now, while its variability remains rather similar.

Numbers have been updated throughout the manuscript based on these new analysis. Also note that Fig. 8 in former version of the manuscript has been moved to the Supplementary Material (now labelled Fig. S8).

With respect to the request from Referee 2 for a more careful assessment of the overflow dynamics, we have added a new Figure in the manuscript (Fig. R4 below, and Fig. 9 in the new version of the manuscript) that evaluates in more detail the overflows in the Denmark Strait. Thus, upward and downward transports have been computed by considering the distance from the sill (see triangle in Fig. R4A) in the upper and lower DSO regions ($DSO \uparrow$ and $DSO \downarrow$ respectively). These results demonstrate that, on average, water masses upwell as they move southward approaching the sill and sink once they move away from the sill (Fig. R4D). It helps to clarify the upward and downward vertical transport shown in Figure 8E. This figure also shows that results are robust for the defined Denmark Strait region as the grid points that contribute most to sinking are within the first 150 km closest to the sill (Fig. R4D), with little change in the amount of sinking in the subsequent 100 km. Regarding the Iceland-Scotland region, we note that the two sills are still present in Fig. R1F (Fig. 8F in the new version), one at around 80 km from the Faroe Islands (closer to Scotland) and another one at 180 km from the Faroe Islands (closer to Iceland). We note that some modifications on the origin of accumulated have been applied following suggestions of this Referee (as well as from major point 1 of Referee 1), which are further explained below in the major point 3. To conclude, the discussion of Fig. 9 has been included in Sect. 4.2 (Page 19, Ln 17-26) as follows:

*"The positive and negative accumulated $W_\Sigma$ in the Denmark Strait over the first 250 km off $C_{50}$ (Fig. 8E) reflect waters moving southward from the Nordic Seas that first flow up and then down over the sill. This is illustrated by the deepening of the isopycnal in Fig. S6C after crossing the Denmark Strait and demonstrated in Fig. 9, in which the Denmark Strait region has been divided in two parts of similar size on either side of the sill (green triangle in Fig. 9A): one that mainly contains the upward movement of waters as they approach the sill (DSO ↑) and another that contains the downward movement of waters after crossing the sill (DSO ↓). As a result, this up/down transport is clearly reflected in Fig. 9B-C, with the strongest upwelling (+1 Sv) located at a depth of 579 m, and the strongest sinking (−3 Sv) is found at 729 m. The difference between DSO ↑ and DSO ↓ accounts for the near 2 Sv of net sinking found in this region. Furthermore, the accumulated vertical transport with respect to the distance to the sill at the respective depths of strongest upwelling and sinking show that the most important contributions occur within the first 150 km off the sill."*
* * *
**2. The overturning circulation: The overturning circulation shown in figure 1A has several features which may not be realistic - e.g. the large decrease in stream function north of 35N, the very shallow depth, particularly north of 45N. See Dunne et al, 2012 J.Climate, figure 4 for examples of overturning which extend further north and deeper. What connection might there be between the diagnosed pattern of overturning, and the dominant horizontal and vertical locations of downwelling seen in this simulation?**

■ **Reply 2.** Despite it is true that the AMOC is not as realistic as it could be, this simulation has some reasonable key features that are further discussed in Section 2 of the manuscript: such as the magnitude of transport in the subpolar Atlantic, the depth of strongest AMOC, its range of variability and the phase of its seasonality. However, in our view the most important point here is that in most simulations and in the observations (RAPID and OSNAP arrays) there is a decrease in the transport of the AMOC between mid-latitudes (30-40°N) and after 45-50°N (also in Dunne et al. (2012), Fig. 8B-C therein). This decrease, by mass conservation, has to be reflected in the amount of sinking in the subpolar region and, although the magnitude of sinking may change between simulations, its regional assessment provides important insights on their distribution and on the physical processes driving it. To state this more clearly, the beginning of Section 3 (Page 6, Ln 25-28) has been modified as follows:

*"The structure of the AMOC streamfunction (Fig. 1A) indicates that there is a decrease in the amount of transport between the North Atlantic mid-latitudes and the subpolar region that, by mass conservation, must be reflected in the magnitude of the vertical transport. However, such figures only provide a two-dimensional view of the overturning circulation*

*in the subpolar North Atlantic. In this study we analyze the complex full structure of the circulation by characterizing spatial and seasonal variations in the sinking."*

Regarding the Referee's last question on the connection between sinking and the AMOC that we further discuss in Section 5 of the manuscript; with the present approach we cannot establish a clear connection between the AMOC and the sinking in the sense that we cannot predict how a regional change in sinking will affect the AMOC. For this reason we propose that a Lagrangian assessment can be more suitable to bring more light on this relationship, since it can help to understand pathways of water masses and time scales involved in the variability of boundary sinking and how they propagate to other regions. We are working on this approach at this moment and we expect to show our results in the coming months. ───────────────

**3. Choice of aggregation by distance from the coast: For the overflows in particular, but also for some other regions (e.g. wide shelves compared to narrow shelves), I wonder if there might be a better method of aggregating the downwelling than by distance from the coast. For example, would bathymetric depth be more physically meaningful than distance from the coast? Figure 4 inset shows that distance from the coast gives undue influence to Jan Mayen island in the Greenland sea, for example. If the downwelling occurs next to the topography, we might expect to see a close relationship between the depth of maximum downwelling and the bathymetric depth, which is hidden when aggregating by distance from land.**

■ **Reply 3.** Also following a similar question from Referee 1 (major point 1), we have reduced the undesired effect of the shelf in Fig. 4 and Fig. 8 of the manuscript by accumulating sinking with respect to the closest contour of 50 meters of depth ($C_{50}$) in turn of the coastline (see Fig. 4 and Fig. 8 in the revised version of the manuscript or Fig. R1 here). This modification, together with the redefinition of the eastern regions (specially the ISO region) and the inclusion of Fig. R4 (Fig. 9 in the new version of the manuscript) helps to clarify the overflow dynamics. Following the Referee's suggestion we have also tried to accumulate sinking using other approaches, e.g. with respect to the bottom depth. Although consistent with the magnitude of sinking, these other approaches are significantly harder to explain (and sinking regimes more difficult to visualize) than the simpler horizontal vision shown in Fig. R1.

The connection between sinking and topography is highlighted in Fig. R3, where the bottom depth is plotted against the depth of largest sinking for the whole domain and for individual regions (defined in Fig. R2). We note that only those grid points with intense sinking ($W_{min} < -0.02$ Sv are shown. As suggested by the Referee, a close linear relationship is evident between topography and sinking, remarkably in the Labrador, Irminger and Newfoundland Seas. Besides there are some misaligned points on the right side of Newfoundland that we associate with sinking within the interior eddies (see a more detailed response on Newfoundland eddies in the reply to the short comment, SC1). We note that Fig. R3 has not been added to the manuscript nor the supplementary material since a similar information can be deduced from Fig. 2B, in which the depth of the near-boundary strongest sinking is very close to the bathymetric contours. Also the new Fig. S9, discussed below, provides information in the same direction in the Labrador and Irminger Seas.

**4. Interpretation of upwelling: I find it hard to interpret the upper ocean/near boundary upwelling seen in the Irminger sea (figure 7B and 9B), and the near boundary upwelling seen in the Denmark Straits (figure 9E). What does this imply physically - how is the circulation closed, where is the upwelled water going? Would a section, showing velocities binned by horizontal and vertical distance, help provide more information?**

■ **Reply 4.** The comment on the Denmark Strait has been addressed in major point 1, so we now focus on the Irminger Sea. To this we have selected a section across the Irminger Sea (Fig. R5, see location of section in the inset within panel A), where the shading color indicates the perpendicular component of velocity ($v$) and we have depicted the tangential velocity and the vertical velocity ($u$, $1000 \times w$) field by black arrows. Overall this figure shows an offshore component of $v$ (indicated by the dominant blue color, i.e. waters move out of the paper) suggesting that the bathymetry is pushing the water off the slope) as well as a persistent upward and downward vertical movement of waters during the all year that shows little change in intensity (compare A-D panels). The vertical flow seems to be mostly driven by bathymetric features since the transport is more intense near the bottom. Interestingly, these vertical displacements of water may propagate long distances upward (>1000 m) even sometimes reaching almost the surface layers thus suggesting a rather barotropic behaviour of the water column. We note that Fig. R5 has not been included in the manuscript nor in the supplementary material because we have included Fig. S9, which is described in the reply to major point 5 and complements this reply on the Irminger Sea upwelling.

**5. Small scale motion of alternating sign: In figure 2A, I have difficulty seeing a clear signal of net downwelling in any region around the northern boundary because of the large amplitude signals of alternating sign. Whereas the mean Labrador sea signal in figure 7 is for near-boundary downwelling, the mean Irminger sea signal is for near-boundary upwelling, yet in figure 2A, the two regions look qualitatively similar, with large amplitude alternating signals. Is it possible to apply a spatial filter to the fields in figure 2A, to show what the vertical motion looks like when averaged over the eddy scales? Its also not clear to me why the Irminger sea alternating signal is suggested to be tied to topography - are there features in the bathymetry on these scales? If so, aggregating by bottom depth might be a way to clarify.**

■ **Reply 5.** We agree with the Referee. The resolution of Fig. 2 of the manuscript is not good enough to appreciate all these subtleties. To clarify this point a new figure considering the vertical transport in the Labrador and Irminger Seas has been made (Fig. R6). First, the mean $W$ at a depth of 1139 m is shown in the near-boundary region of Labrador and Irminger Seas with shading color. It allows to see clearer the coherent patterns of up and down vertical transport. Next, over the bathymetric contour depth of 1800 m (see gray contour line in the main panel) we have estimated the mean vertical transport, $W$ at a depth of 1139 m (shading color). Then we have plotted it starting in the Labrador Sea and finishing in the Irminger Sea (see inset in Fig. R6). The separation between both seas according to the regional map shown in Fig. R2 is also denoted by a dashed vertical dark red line. This plot shows the main differences in sinking patterns between both seas, which are characterized by a more erratic shape in the Labrador Sea and a clearer up and down movement in the Irminger Sea, specially at distances >600 km. It also helps to stress (together with Fig. R5) the key role that bathymetric features play in the Irminger Sea. For completeness this figure has been added in the Supplementary Material as Fig. S9 and is referred in Sect. 4.4 as follows:

> *In contrast to the Labrador Sea, the change in $W$ in the Irminger Sea between the months of minimum and maximum time-mean $W_\Sigma$ is significantly smaller (Fig. 10B). This finding, together with the permanent depth of largest sinking ($z_{\min}$) and the up and down distribution of sinking found in Fig. 8B, supports the hypothesis that the sinking near the boundary in the Irminger Sea is mostly quasi-stationary and topographically-driven (see a detailed example of this up/down of waters in Fig. S9), which explains the small amount of net sinking found.*

[Figure]

Figure R4: (A) Map of Denmark Strait Overflow region (DSO). The mean vertical transport ($W$) at 729 m is depicted by shading (color). The triangle illustrates the location of the sill (green triangle) that separates the Denmark Strait in two areas of similar size (DSO ↑ and DSO ↓). Bathymetric contours are indicated by black line. (B-C) Vertical structure of transport ($W_\Sigma$) in DSO ↑ and DSO ↓ respectively. (D) Annual (dashed line) and monthly (red solid lines) accumulated vertical transport with respect to the sill in DSO ↑ (red) and in DSO ↓ (light blue ). Both have been calculated at the corresponding depths of largest upwelling (579 m) and sinking (729 m) for DSO ↑ and DSO ↓ respectively.

[Figure]

Figure R5: 15-year climatology of the velocity field at a cross-section between the southern tip of Greenland and the southern limit of the study area (see inset in panel A). Each panel represents a seasonal average: (A) JFM (January-February-March); (B) AMJ (April-May-June); (C) JAS (July-August-September); OND (October-November-December). The shading shows the normal velocity component (indicated by $v$), with units in $m\,s^{-1}$; black arrows are velocity vectors constructed as $(u, 1000 \cdot w)$ where $u$ and $w$ are the tangential and the vertical velocity components respectively. For clarity arrows are shown for selected depths and grid points. The green line depicts the seasonal mean mixed layer depth (in m), while the black contours denote the seasonal potential density anomaly, $\sigma_\rho = \rho - 1000$.

[Figure]

Figure R6: Mean vertical transport ($W$) at a depth of 1139 m (shading color) in Sv. The grayscale contour indicates the bathymetric contour of 1800 m. Its grade of colors denotes the distance over this contour from southwestern Greenland to southeastern Greenland in km. The inset panel shows $W$ at the those grid points located at the contour of 1800 m at a depth of 1139 m according to the distance defined in the main panel. The vertical dark red dashed line marks the boundary between Labrador and Irminger Seas according to Fig. 6.

**Specific comments:**

**Page 2, Line 3: Insert "few" after "Over the last"**

■ **Reply 6.** Done. ─────────────────────────────────────

**Page 2, Line 31: Change "looses" to "loses".**

■ **Reply 7.** Thank you. This typo has been corrected. ─────────────────

**Page 3, Line 28: Change "and idealized eddy-resolving model" to "an idealized eddy-resolving model".**

■ **Reply 8.** Done. ─────────────────────────────────────

**Page 4, line 3-4: "not adding even more complexity" - Im not sure what you mean by this, perhaps not adding the additional complexity of historical variations in surface forcing?**

■ **Reply 9.** As suggested we have redone this unclear statement. Now it reads:

> *To this end, we use an ocean-only eddy-resolving numerical simulation with a nominal resolution of 0.1° under a repeated climatological annual atmospheric forcing, not adding the additional complexity of historical variations (e.g. at inter-annual or inter-decadal scales) in surface forcing.*

[Figure]

**Page 6, line 20-21: Change "Nevertheless it can be inferred a stronger transport in winter than in summer" to "Nevertheless, a stronger transport in winter than in summer can be inferred . . . "**

■ **Reply 10.** Done. ─────────────────────────────────────

**Figure 2B: The depth color bar shows increasing depth upward, whereas it is more intuitive to show it downward on the color scale (a minor point, I know)**

■ **Reply 11.** The colorbar has been reversed accordingly. ─────────────────

**Figure 3: Given this downwelling in Sv, if you divided through by the area at each depth, what would this imply for the magnitude of the vertical velocity in m/s? How does this compare with the magnitudes shown in figure 2A? (Im guessing that the noise shown in figure 2A is much greater than the mean signal).**

■ **Reply 12.** For the Labrador Sea case, the surface at the depth of 1139 m is around $7 \cdot 10^6 \text{km}^2$. Then, for a total downwelling of 4 Sv the mean vertical velocity is around 0.5 m/day. In this regard we note that we are using monthly mean fields and therefore the strong velocities found in the boundary respond to some averaged physical processes. Additionally, the up and down pattern of vertical transport shown in Fig. R6 appear to be rather spatially coherent and not isolated spots. Additionally the values of standard deviation in Fig. 2C are rather smaller (around 5 times) than the near boundary vertical velocities shown in Fig. 2A. Therefore we are convinced that the near-boundary sinking is not noise. It has been explicitly stated in Page 7 Lines 8-10 as follows:

*"These strong vertical velocities velocities cannot be considered as noise since they show a coherent spatial pattern along the bathymetric contours and their standard deviation is several times smaller (square root of variance shown in Fig. 2C, about 30 m day 1 ) than their mean value."*
* * *
**page 10, line 4-5: If the W value is shown at a depth of 1139 m, what do you do if the grid cell bathymetry is shallower than that value?**

■ **Reply 13.** Then it is assumed that there is no value to be added. It has been explicitly stated in the caption of Fig. 4:

*"Cumulative net vertical transport ($W_\Sigma$) at a depth of 1139 m (in Sv) as a function of the distance from the bathymetric contour of 50 m depth referred to as $C_{50}$ (inset map in Fig. 4). If the grid cell bathymetry is shallower no value is added. The dashed black line shows the annual mean. Monthly values of the 15-year simulation are shown in light gray, colored lines indicate the monthly climatology. The regimes of sinking are indicated by roman numbers **I-II-III**, and the separation lines between them are also denoted by a brown and a black triangle and by contours in the same color in the inset figure."*
* * *
**p11 discussion of figure 5, and figure 5: To see the context of the currents around 1250km-1000km south of Greenland, it would be helpful to show an inset of the surface currents over the whole North Atlantic. How is this "standing eddy" related to the North Atlantic Current?**

■ **Reply 14.** We have added the main surface features around the section in the inset of Fig. 5 of the manuscript. However, due to the small size of the inset for a complete view of the mean North Atlantic circulation we refer to Fig. S1 (top panel) in the Supplementary Material. Also Fig. R7 (labelled Fig. S4 in the supplementary material) provides a good view of the currents near the Flemish Cap.

Regarding the eddies, large eddies are detached from the North Atlantic current as it meanders on their way to the North (see Fig. R7). These eddies move around the Flemish Cap and few of them even propagate northwestward into the central Labrador Sea. Indeed they can stay for months nearby the Flemish Cap region before dissipating. The big red/blue patches of W in Fig. 10 of the manuscript are a good indicator of the vigorous eddy field present in Newfoundland and of their role on the marked upward/downward motion that we find there. ─────────────

**page 26, lines 30-31: Change "limit us to find" to "prevents us from finding".**

■ **Reply 15.** Done. ────────────────────────

**page 27, line 5: Change "isopycnals significantly fluctuate" to "isopycnals fluctuate significantly"**

■ **Reply 16.** Done. ────────────────────────

**Short Comment (SC1) from Damien Desbruyères**

**I have a question regarding the particular region of Flemish Cap, where you report a relatively important eddy-driven sinking. I wonder how the latter reconciles with the key ingredient for vertical sinking: a positive along-stream density gradient supporting an in-shore geostrophic flow in the upper layer. Through mass conservation this drives downwelling along the slope, as the authors well explained in their introduction. This is not the case of the boundary current encircling Flemish Cap, which is warming as it flows downstream due to exchanges with the warmer North Atlantic Current, so that the sign of the along-stream density gradient would rather suggest upwelling there. Could the authors comment on this?**

■ **Reply 1.** First of all thank you for your interest in our work. We think that the term "eddy-driven sinking" (or eddy-induced, a term that we also use) is ambiguous, since it could be used for any of the sinking regimes (**I-II-III**). Surely it has given place to an inaccurate interpretation of our words. The reason in that in the near-boundary sinking (at around 1000 m, closer to the shelf) as well as in the offshore sinking (that sometimes extends below 2000 m and is found farther offshore), eddies play a central role although in a very different dynamical way.

In the following lines we discuss the dynamics around and off the Flemish Cap in order to bring more light on the dynamical processes occurring there according to our model simulation. First, this region is characterized by a strong front, as indicated by the contours of potential density anomalies (black thick contours) in Fig. R7A-B, in which A-B panels correspond to two different depths (FC refers to Flemish Cap). The strength of this front is illustrated by the decrease in about $0.6$ kg $\cdot$ m$^{-3}$ in $\sim 200$ km when we move off the eastern side of Flemish Cap . However, below 1000 meters the potential density is rather homogeneous around the Flemish Cap, with almost constant values of $28.01 - 28.02\, kg \cdot m^{-3}$ at 2125 m (some insight can be inferred from Fig. S6C in the supplementary material) that change little along the current pathway (see trajectory in Fig. R7C-D). Based on this, a major relevance of other sinking mechanisms different from the near-boundary sinking is expected to explain the strong vertical velocities seen (shading in Fig. R7).

Indeed from Fig. R7C-D we can infer that as the current approaches to the Flemish Cap, it first moves up as it faces shallower bathymetric contours (see the dominant red patch in the northwestern Flemish Cap in Fig. R7C-D), and later moves down as water finds a deeper bathymetry (see the subsequent blue patch). This explains the up/down closer to the Flemish Cap which should be roughly compensated to provide little net transport (as it is indicated by the small sinking in the first 200 km found in Fig. R1G or Fig. 8G in the manuscript).

However, if now we move a bit farther off the Flemish Cap to the interior of Newfoundland, also large blue and red patches appear at the edges of mesoscale eddies (indicated by the enclosed black arrows). In this case, the upward/downward movement of waters is governed by the own eddy dynamics, which is also different from near-boundary sinking. The reason of this sinking at the edges is unclear for us and will require further study, which is not attainable with the monthly fields used here. Some works have mentioned the fast vertical displacement of isopycnals as one plausible cause for strong vertical velocities of mesoscale geostrophic eddies, e.g. Viúdez and Dritschel (2003). Therefore, if near-boundary sinking exists in this region, it is rather limited and hardly comparable to the one in the Labrador Sea, as suggested by Fig. R1G (Fig. 8G in the manuscript).

We hope that this explanation will clarify more the processes occurring within the Newfoundland region, as well as the processes that can be seen in Fig. 5 of the manuscript. Fig. R7 has been added to the supplementary material as Fig. S4. In the new version of the manuscript we have avoided the expression eddy-driven (eddy-induced) sinking when we refer to the sinking that occurs farther offshore at the edge of large interior eddies (regime **III**). The main changes are indicated below.

- In Sect. 3.1 (Page 8. Ln 1-6):

  *"The positive and negative alternations offshore of the Flemish Cap and in the interior of Greenland and Norwegian Seas must have a different cause. In the case of the Flemish Cap, they occur at the edges of eddies (Fig. S4) and the depth of largest sinking is below 2000 m (Fig. 2B, also in the interior of Norwegian and Greenland Seas), which indicates these eddies are deep and possibly have a strong barotropic component. Indeed, the high variance of vertical velocities in the surroundings of the Flemish Cap ($\sigma^2(w)$) is a reflection of the existence of an active eddy field throughout the year (Fig. 2C). Also the subsurface EKE shows this signal (Fig. S5)."*

- The last paragraph of Sect. 3 (Page 13, Ln. 19-23 ) has been rewritten as follows:

  *"Finally, the sinking in regime **III** is related to those processes that develop away from the shelf and far from the core of the boundary current. In this case, strong vertical velocities appear at the edge of interior eddies, which are governed by a different dynamics (Fig. 5). The major role of such quasi-permanent eddies is supported by the marked fluctuations between 300 and 1000 km in Fig. 4, the large interior eddy in Fig. 5 and the vigorous EKE field in the interior of the Newfoundland Basin, near the Flemish Cap (Fig. S4 and Fig. S5)."*

- The last paragraph in Sect. 4 (Page 24, Ln. 4-12) has been rephrased to avoid any ambiguity:

  *"Newfoundland yields the second largest contribution to sinking, which is largely produced within the sinking regime **III**. The strong seasonal variations of W in the Newfoundland region below 2000 m are mostly related to EKE interior pathway changes (see black contours in Fig. 10B) impinged by North Atlantic Current fluctuations and meandering. The fact that strong vertical velocities appear at the edge of large eddies in the interior, thus contributing to upwelling/sinking has been already shown in Fig. 9G, Fig. 10A-B and Fig. S4, where a train of large eddies near the Flemish Cap is visible). Finally, in the Newfoundland region the peak of the boundary current at its corresponding $z_{\min}$ falls in regime **III** (Fig. 11E); although the strongest sinking occurs in the interior and below 2000 m, there is some sinking at shallower depths as indicated by its vertical structure in Fig. 7G (Fig. S4). Similar plots showing the overall weaker spatial patterns of sinking for the remainder of the regions can be found in Fig. S10 and Fig. S11."*

[Figure]

Figure R7: Mean vertical velocity at different depths around the Flemish Cap ($\overline{w}$): (A) depth of 130 m, (B) depth of 466 m, (C) depth of 1379 m, (D) depth of 2125 m. Thin contours represent bathymetry in all panels. Black thick contours in (A-B) represent the potential density anomalies. Black arrows in (C-D) denote the horizontal current fields. For all cases the 15-year mean is shown. FC refer to the Flemish Cap.

[revised manuscript text omitted]